# DNA lesions can frequently precede DNA:RNA hybrid accumulation

Raphaël M. Mangione [1], Steven Pierce[2,8], Myriam Zheng [1,2,8], Robert M. Martin [3,4,8], Coralie Goncalves[1], Arun Kumar[5,6], Sarah Scaglione [7], Cristiana de Sousa Morgado[3,4], Arianna Penzo [1], Astrid Lancrey[1], Robert J. D. Reid [2], Ophélie Lautier[1], Pierre-Henri Gaillard [7], Peter C. Stirling [5,6], Sérgio F. de Almeida [3,4], Rodney Rothstein [2] & Benoit Palancade [1] ✉

While DNA:RNA hybrids contribute to multiple genomic transactions, their unscheduled formation is a recognized source of DNA lesions. Here, through a suite of systematic screens, we rather observed that a wide range of yeast mutant situations primarily triggering DNA damage actually leads to hybrid accumulation. Focusing on Okazaki fragment processing, we establish that genic hybrids can actually form as a consequence of replication-born discontinuities such as unprocessed flaps or unligated Okazaki fragments. Strikingly, such "post-lesion" DNA:RNA hybrids neither detectably contribute to genetic instability, nor disturb gene expression, as opposed to "pre-lesion" hybrids formed upon defective mRNA biogenesis, *e.g.*, in THO complex mutants. Post-lesion hybrids similarly arise in distinct genomic instability situations, triggered by pharmacological or genetic manipulation of DNA-dependent processes, both in yeast and human cells. Altogether, our data establish that the accumulation of transcription-born DNA:RNA hybrids can occur as a consequence of various types of natural or pathological DNA lesions, yet do not necessarily aggravate their genotoxicity.

RNA-containing structures represent an increasingly recognized fraction of genomes, from unicellular organisms to metazoans[1]. RNA moieties are incorporated into DNA duplexes during replication, as ribonucleotide insertions or DNA:RNA hybrids, i.e., those priming the synthesis of Okazaki Fragments (OF), and during transcription, in the form of R-loops, when the nascent transcript anneals back to its template and displaces the non-template strand as a single-stranded DNA (ssDNA)[2]. While such hybrid structures are natural intermediates in several DNA-related transactions, their unscheduled accumulation jeopardizes genome homeostasis[3,4]. Accordingly, a myriad of factors

and processes have been reported to regulate the turnover of DNA:RNA hybrids and R-loops[5]. For instance, RNA primers inserted during OF synthesis are removed either by the hybrid-specific ribonuclease RNase H2 or through displacement by DNA polymerases, followed by cleavage of the resulting 5′-flaps by structure-specific endonucleases (e.g., scRad27/hsFEN1) or exonuclease activities, prior to sealing by DNA ligase I[6,7]. The metabolism of R-loops is similarly kept in check by a series of players, most of them identified in yeast or human cells through genetic screens for factors preventing genome instability, or proteomic screens for hybrid-associated proteins[8–10].

[1]Université Paris Cité, CNRS, Institut Jacques Monod, Paris, France. [2]Department of Genetics & Development, Columbia University Irving Medical Center, New York, NY, USA. [3]GIMM—Gulbenkian Institute for Molecular Medicine, Lisbon, Portugal. [4]Faculdade de Medicina da Universidade de Lisboa, Lisbon, Portugal. [5]Terry Fox Laboratory, BC Cancer, Vancouver, BC, Canada. [6]Department of Medical Genetics, University of British Columbia, Vancouver, BC, Canada. [7]Centre de Recherche en Cancérologie de Marseille (CRCM), U1068 Inserm, UMR7258 CNRS, Institut Paoli-Calmettes, Aix Marseille Université, Marseille, France. [8]These authors contributed equally: Steven Pierce, Myriam Zheng, Robert M. Martin. ✉e-mail: benoit.palancade@ijm.fr

                                                    

R-loop formation is notably prevented through proper RNA processing, packaging with RNA-binding proteins (e.g., the THO/TREX complex), and nuclear export[11], while R-loop clearance further involves ribonucleases of the RNase H family, together with DNA:RNA helicases (e.g., Sen1/SETX), some of them engaged during transcription, DNA replication or DNA repair[12–15].

Harmful hybrids have been reported to inhibit transcription elongation and to trigger transcription-associated genetic instability, enhancing the rates of mutations, DNA double-strand breaks (DSBs), and genome rearrangements[5]. The genotoxicity of R-loops can notably stem from the recognition of their ssDNA moieties by nucleases or DNA-modifying enzymes, and from their propensity to trigger replication stress and transcription-replication conflicts[4]. However, systematic mapping of hybrid positions within eukaryotic genomes, either by using the hybrid-specific S9.6 monoclonal antibody, or by taking advantage of their recognition by RNase H hybrid-binding domains, has failed to fully elucidate the determinants of their genotoxicity[16]. Indeed, not all DNA:RNA hybrids or R-loops are associated with DNA damage and replication stress features, suggesting that they are not equally genotoxic[17,18]. Furthermore, situations of artificially-induced DSBs[19–24] or replication stress[25–28] have themselves been associated with increased local levels of DNA:RNA hybrids or R-loops, raising the question of the temporal relationship between the formation of such hybrid structures and their associated DNA lesions.

## Results

### Systematic screens reveal novel sources of DNA:RNA hybrids

To systematically investigate the relationships between DNA:RNA hybrid levels and genetic instability, we designed a suite of screens to simultaneously score both phenotypes in yeast mutant libraries. In the first genome-wide screen, we built a plasmid construct in which the *LYS2* gene, a sequence prone to DNA:RNA hybrid accumulation and hybrid-dependent genetic instability[29–31], is inserted between overlapping fragments of the *YFP* gene, which encodes the yellow fluorescent protein (Fig. 1a, screen I). In such a configuration, transcription-dependent DNA:RNA hybrid or R-loop formation likely enhances direct-repeat recombination by single-strand annealing (SSA)[32], allowing the reconstitution of a functional reporter gene, a potential proxy for hybrid-dependent genetic instability. A high-throughput plasmid transfer procedure[33] (Supplementary Fig. 1a) was used to introduce the reporter construct into the *S. cerevisiae* knock-out library, which encompasses all deletants for non-essential genes. YFP expression levels were then measured in all individual mutants by flow cytometry. Multiple sampling and comparison to wild-type distributions allowed us to define recombination indices (see "Methods") and to rank 4460 mutant strains according to their divergence from controls (Fig. 1b). Using this ranking, a cutoff based on interaction knowledge[34] defined a sub-library of 114 mutants that trigger hyper-recombination (Fig. 1c and Supplementary Data 1).

To confirm their phenotype, mutants from the primary screen were subsequently transformed with an alternative SSA reporter expressing the *YAT1* gene, a distinct R-loop forming sequence[29,30,35]. Hyper-recombination was then similarly scored by assessing the reconstitution of the *LEU2* auxotrophy marker in high-throughput replica pinning assays[36] (Fig. 1a, screen II). This secondary screen ultimately identified 39 genes whose deletion triggered a significant hyper-recombination phenotype on both R-loop-forming reporters (Fig. 1d and Supplementary Data 1). Candidate genes were mostly conserved from yeast to human (Supplementary Fig. 1b) and impacted various biological processes, including DNA repair, recombination, replication, transcription by RNA polymerase II (RNAP II), chromatin organization, and more unexpectedly, mitochondrial components, as revealed by gene ontology analysis (Fig. 1e and Supplementary Fig. 1c). Importantly, both screening assays show a strong genetic dependency on *RAD52* and increased recombination in mutants of the bona fide

hybrid-preventing THO complex[37] (Fig. 1f, g and Supplementary Data 1). More generally, these successive screens identified a specific set of hyper-recombinant mutants compared to published SSA screens based on reporters that do not carry any R-loop forming sequence[36,38] (Supplementary Data 1), thus highlighting our potential to capture R-loop dependent events.

To further assess hybrid accumulation in these hyper-recombinant mutants, their genomic DNA was probed in dot blot assays with the S9.6 antibody (Fig. 1a, screen III and Supplementary Fig. 1d), whose specificity for hybrids was validated by in vitro RNase H treatment (Supplementary Fig. 1e). As expected, a large fraction of these candidate mutants, including those in the THO complex, accumulated genomic hybrids as compared to control strains, in proportions comparable to the prototypical RNase H double mutant, *rnh1Δ rnh201Δ* (Fig. 1h and Supplementary Fig. 1f). Strikingly, mutants impacting OF processing, i.e., *rad27Δ* or *cdc9**, exhibited amongst the highest levels of DNA:RNA hybrids in the screen (Fig. 1h–i). In parallel, we assessed the consequences of OF processing defects in human cells by using pharmacological inhibitors of FEN1, the Rad27 ortholog[39,40]. Live imaging assays using fluorescent sensors derived from the RNase H1 DNA:RNA hybrid-binding domain[41] revealed specific focal signals in MCF7 breast cancer cells (Supplementary Fig. 2a), which were abolished when hybrid recognition was compromised (W34A and KK59/60AA mutants; Supplementary Fig. 2b). Strikingly, treatment with two distinct FEN1 inhibitors led to enhanced hybrid signals (Fig. 1j and Supplementary Fig. 2a, c), similar to what we observed in yeast.

Although our systematic strategies thus identified a large number of situations that enhance both recombination and DNA:RNA hybrid levels, the case of OF processing defects, as observed in both yeast and humans, was particularly intriguing. In fact, it is unlikely that accumulating hybrids cause the genetic instability in OF processing mutants, where discontinuities in the newly synthesized lagging strand, i.e., unprocessed flaps or unligated OFs, are the main recombination substrates[6,42–47]. Thus, we next explored the timing of hybrid accumulation upon defects in OF processing.

### Okazaki Fragment processing prevents widespread DNA:RNA hybrid accumulation on transcribed genes

To further characterize the primary source of DNA:RNA hybrid accumulation in OF processing mutants, Rad27 was acutely depleted using the Auxin Inducible Degron (AID) system (Fig. 2a, b). The growth of *RAD27-AID* cells was virtually indistinguishable from *wt* in untreated conditions, while the addition of auxin markedly triggered temperature and MMS (methyl methane sulfonate) sensitivity (Supplementary Fig. 3a), phenocopying *rad27Δ* cells[48,49], and validating the use of this conditional allele for further functional assays. Confirming our screen data, Rad27 depletion led to the accumulation of DNA:RNA hybrids (Fig. 2c, d), which were sensitive to in vitro RNase H treatment (Supplementary Fig. 3b). Importantly, chromosome spreads[50] obtained from fixed *rad27* mutant cells similarly scored a gain of hybrids, which was suppressed by in vivo expression of RNase H1 (Fig. 2e and Supplementary Fig. 3c), ruling out in vitro formation during DNA extraction, as previously proposed[51]. To further examine hybrid distribution at the genome-wide level, *RAD27-AID* cells were used for strand-specific DNA:RNA hybrid immunoprecipitation (IP) coupled to sequencing (DRIP-seq)[52], using *C. glabrata tho* cells[29] as a spike-in for calibration[53] (Fig. 2f). Biological replicates were highly correlated (Supplementary Fig. 3d) and, for the *minus auxin* condition, in agreement with available hybrid maps obtained from *wt* cells[54,55] (Supplementary Fig. 3e). In untreated *RAD27-AID* cells, immunoprecipitated samples displayed RNase H-sensitive peaks (Fig. 2g), which correlated with transcription levels (Supplementary Fig. 3f), as previously observed[55]. These signals were mostly restricted to the template strand of transcription units and excluded from intergenic regions, as highlighted by heat map analysis of the most

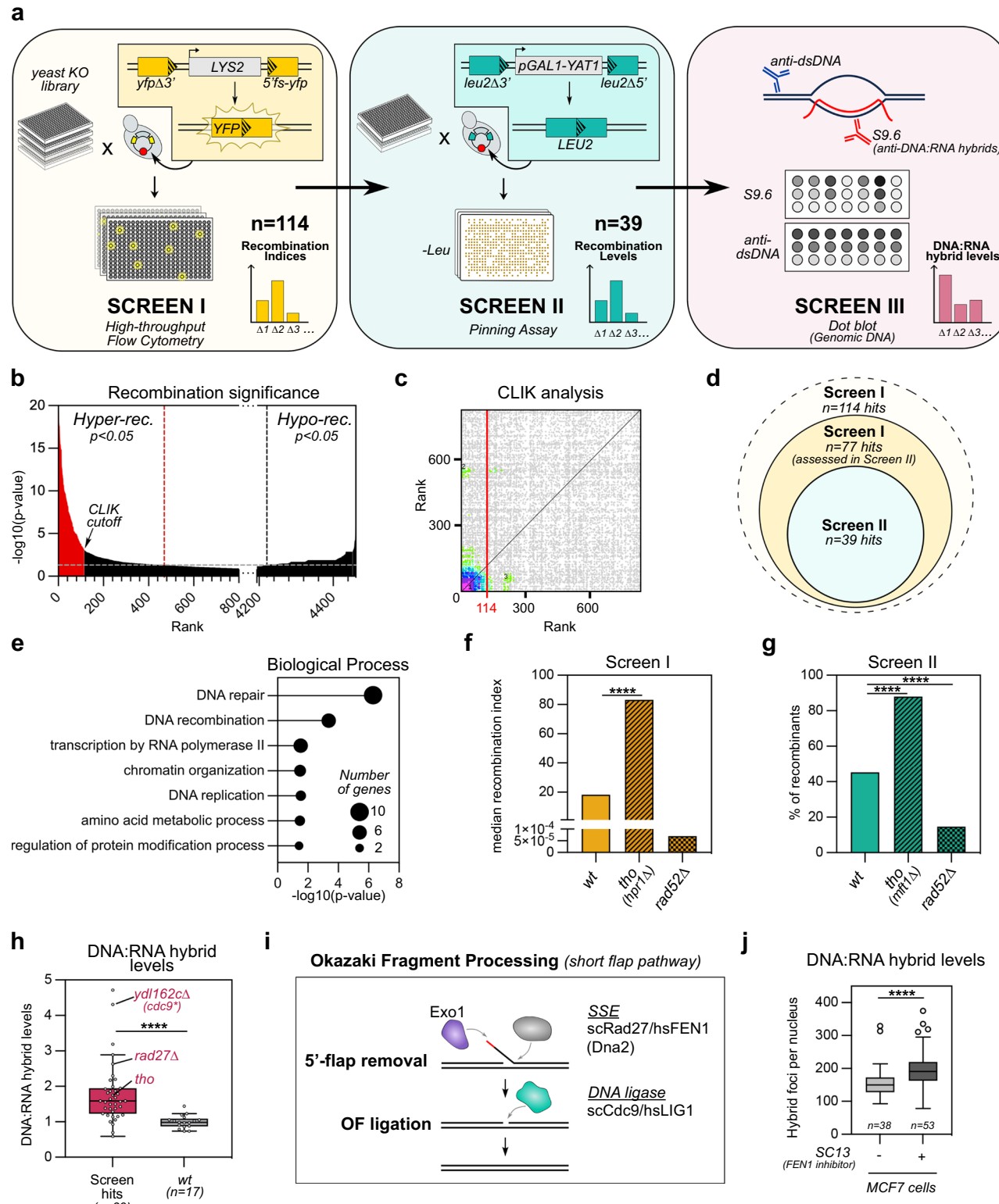

highly-transcribed genes ($n = 170$), ranked according to hybrid levels (Fig. 2h). Together with the fact that this strand specificity was observed throughout the entire genome (Fig. 2i), we conclude that our assays mostly detect transcription-dependent hybrids. Strikingly, Rad27 depletion (plus auxin) led to a widespread increase in hybrid signals at pre-existing hybrid-prone genes (Fig. 2g), without altering either their exclusion from intergenic regions (Fig. 2h), or their strand specificity (Fig. 2i), ruling out the possibility that they originated from OF RNA primers.

Consistently, DRIP-qPCR validation experiments detected hybrid accumulation in the absence of Rad27 on typical hybrid-forming loci (i.e., *PMA1*, *YEF3*, *ADH1*) as compared to intergenic regions (Fig. 2j, Asynchronous cells), a phenotype that was complemented by *RAD27* re-expression (Supplementary Fig. 4a). Global inhibition of transcription (Supplementary Fig. 4b) or specific repression of an inducible *YAT1* hybrid-forming reporter (Fig. 2k) confirmed that DNA:RNA hybrids detected in *wt* cells and accumulating in the absence of Rad27 are formed in a transcription-dependent manner.

**Fig. 1 | Systematic screens reveal novel sources of DNA:RNA hybrids. a** Principle of hyper-recombination and DNA:RNA hybrid detection assays in systematic screens. Reporter genes are positioned opposite to the unique plasmid replication origin (red dot) and can be replicated in both directions with respect to transcription. **b** Mutant strains assessed in Screen I ($n = 4460$) were divided along the x-axis according to recombination indices (left, hyper-rec; right, hypo-rec) and ranked according to their divergence from *wt* (*p*-value). Hashed lines indicate cutoff by $p = 0.05$. **c** Interaction densities between top-ranked hyper-recombinant strains from Screen I, as assessed by CLIK. Numbers (1,2,3): interaction densities significantly above background; red line: CLIK cutoff[34]. **d** Comparison between Screen I and Screen II hyper-recombinant hits. Out of 114 mutants identified in Screen I, only those exhibiting normal fitness in galactose induction conditions ($n = 77$) were assessed in Screen II. **e** Gene Ontology analysis for genes whose deletion triggers significant hyper-recombination in both screens I and II ($n = 39$). **f** Recombination indices (Screen I) for control strains (****, $p = 7.0 \times 10^{-7}$).

**g** Recombination levels (% *LEU*+ recombinants, Screen II) for control strains (*wt* vs *tho* ****, $p = 1.9 \times 10^{-33}$; *wt* vs *radS2Δ* ****, $p = 1.4 \times 10^{-18}$). **h** DNA:RNA hybrid levels for hyper-rec mutants common to Screen I and II ($n = 39$; ****, $p < 0.0001$), as assessed in Screen III. The position of *tho* and OF processing mutants is indicated. *YDL162c* deletion overlaps the promoter of the DNA ligase I gene (*CDC9*) and is considered as a hypomorphic *cdc9* allele. **i** Schematic representation of main players in OF processing. SSE: Structure Specific Endonucleases. The OF RNA primer appears in red. **j** Quantification of DNA:RNA hybrid foci in MCF7 breast cancer cells expressing the RHINO sensor and treated with the SC13 FEN1 inhibitor ($n = 3$; total number of cells: control, $n = 38$, SC13, $n = 53$; ****, $p < 0.0001$). For box plots (**h**, **j**), boxes extend from the 25th to 75th percentiles, with the median displayed as a line. The whiskers mark 1.5 times the inter-quartile range of the first or third quartile (Tukey's definition), displaying all the values (**h**) or outliers (**j**) as individual points. Statistical tests: **b, f, h, j** Two-sided Mann–Whitney–Wilcoxon rank sum test; **e** Hypergeometric test; **g** Two-sided Fisher exact test. Source data are provided as a Source Data file.

To assess whether increased hybrids are caused by loss of Rad27 activity during DNA replication, cells were first synchronized in G1 through alpha-factor treatment (Supplementary Fig. 4c), prior to auxin-induced depletion. Strikingly, Rad27 depletion did not induce DNA:RNA hybrid accumulation in G1 in contrast to asynchronous cycling cells, as observed by DRIP-seq (Supplementary Fig. 4d) and validated by DRIP-qPCR (Fig. 2j, G1). To explore this finding, we released alpha-factor arrested cells and monitored hybrid accumulation in early S and late S/G2 phases (30 min and 40 min after release, respectively; Supplementary Fig. 4e). DRIP-qPCR analysis revealed that the hybrid accumulation caused by Rad27 loss coincided with S phase entry (Supplementary Fig. 4f, T30), with a decline in the late S/G2 phase (Supplementary Fig. 4f, T40). Altogether, these results indicate that loss of Rad27 activity during replication leads to widespread hybrid accumulation at transcribed genes.

## Replication-born discontinuities precede the accumulation of DNA:RNA hybrids

We next explored the temporal and causal relationship between recombinogenic DNA lesions and DNA:RNA hybrids, which are both triggered upon defective OF processing (i.e., Rad27 inactivation). On the one hand, loss of Rad27 primarily leads to the accumulation of unprocessed flaps on DNA duplexes[6]; at transcribed loci, such discontinuities could further favor nascent RNA invasion and hybrid formation. Alternatively, since R-loop 3′ boundaries resemble 5′-flaps (Fig. 3a), Rad27 may directly target R-loop structures and participate in their clearance, as previously observed for human FEN1[56] and other structure-specific endonucleases[57], thus impacting their recombinogenic potential.

We first compared Rad27 endonucleolytic activity towards model synthetic 5′-flaps and R-loop substrates through in vitro assays. Recombinant, purified Rad27 (Supplementary Fig. 5a) efficiently cleaved 5′-flaps at their junction, an activity that was undetectable with a catalytic-dead (D129A) version of the protein (Fig. 3b, left panels, compare *wt* to D129A; Fig. 3c). In contrast, while the cleavage of R-loop 3′ boundaries was detectable (Fig. 3b, right panels), it was several orders of magnitude less efficient than for flaps (Fig. 3d, e).

To further assess which of these Rad27 activities is relevant for hybrid metabolism in vivo, we took advantage of a separation-of-function mutant (*rad27-E176A*), described as competent for flap processing, but less proficient towards alternative substrates[58,59]. Since this mutant protein is notably impaired for DNA bubble recognition[59], we hypothesized that it would also be affected by R-loop cleavage. Using in vitro assays, we found that Rad27-E176A indeed failed to process R-loop substrates, while it cleaved flaps as efficiently as the *wt* protein (Fig. 3b–d, compare *wt* to E176A). Strikingly, when expressed in vivo in Rad27-depleted cells, the Rad27-E176A protein fully suppressed DNA:RNA hybrid accumulation, akin to its *wt* counterpart, as assessed by DRIP-qPCR (Fig. 3f and

Supplementary Fig. 5b). This result supports the notion that the accumulation of unprocessed flaps is the primary cause for hybrid gain in the absence of Rad27.

To confirm this finding, we interfered with flap levels by modulating the activity of Exo1, an exonuclease acting in a back-up pathway for flap removal and OF processing[60]. On the one hand, *EXO1* overexpression suppressed the growth defects caused by Rad27 depletion at 37 °C (Fig. 3g), in line with earlier studies[60,61], and also prevented DNA:RNA hybrid accumulation (Fig. 3h and Supplementary Fig. 5c). On the other hand, *EXO1* inactivation led to synthetic sickness in combination with Rad27 inactivation (Fig. 3i), as reported[43,47], and synergistically enhanced hybrid levels in Rad27-depleted cells (Fig. 3j and Supplementary Fig. 5d). Altogether, these results demonstrate that flap accumulation is sufficient to trigger hybrid formation.

The fact that we had also identified a *cdc9* (DNA ligase I) hypomorphic allele in our screens (Fig. 1h) prompted us to directly test whether nicks within DNA templates also lead to DNA:RNA hybrid accumulation. For this purpose, we assessed DNA:RNA hybrid levels in the *cdc9-1* mutant, which has been shown to accumulate unligated OF[62]. Strikingly, *cdc9-1* cells exhibited DNA:RNA hybrid accumulation (Fig. 3k and Supplementary Fig. 5e), a phenotype that was epistatic with the one caused by Rad27 depletion (Fig. 3k and Supplementary Fig. 5e; compare *cdc9-1 RAD27-AID* plus or minus auxin). While we cannot exclude that replicative stress or other replication-dependent damage arising in these genetic contexts could participate in hybrid formation, these data indicate that the accumulation of discontinuities within neosynthesized lagging strands, i.e., unprocessed flaps, or nicks between unligated OF termini, is sufficient to enhance the formation of DNA:RNA hybrids at hybrid-prone transcribed regions.

## Post-lesion DNA:RNA hybrids neither impact gene expression nor genetic stability

Our results support the existence of a class of DNA:RNA hybrids formed as a result of DNA discontinuities, which we term "post-lesion" hybrids, to distinguish them from the well-characterized "pre-lesion" genotoxic hybrids primarily arising from defective mRNA biogenesis, e.g., in mutants of the THO complex. We then compared the impact of both classes of hybrids on genome functions. DNA:RNA hybrids formed in *tho* mutants are typically associated with decreased RNAP II elongation rates, with marked effects on gene expression[63,64]. In contrast, depletion of Rad27 had little effect on steady-state mRNA levels, as revealed by RNA-seq (Fig. 4a). In addition, the few transcripts that were down-regulated did not specifically originate from genes with strong hybrid gain as scored in our DRIP-seq analyses (Fig. 4b). Further analysis of mRNA levels and RNAP II recruitment on the R-loop-forming *YAT1* reporter did not reveal major effects of *RAD27* inactivation on transcription, in contrast with *tho* mutants, as shown by RT-qPCR (Supplementary Fig. 6a) and chromatin IP (Supplementary Fig. 6b).

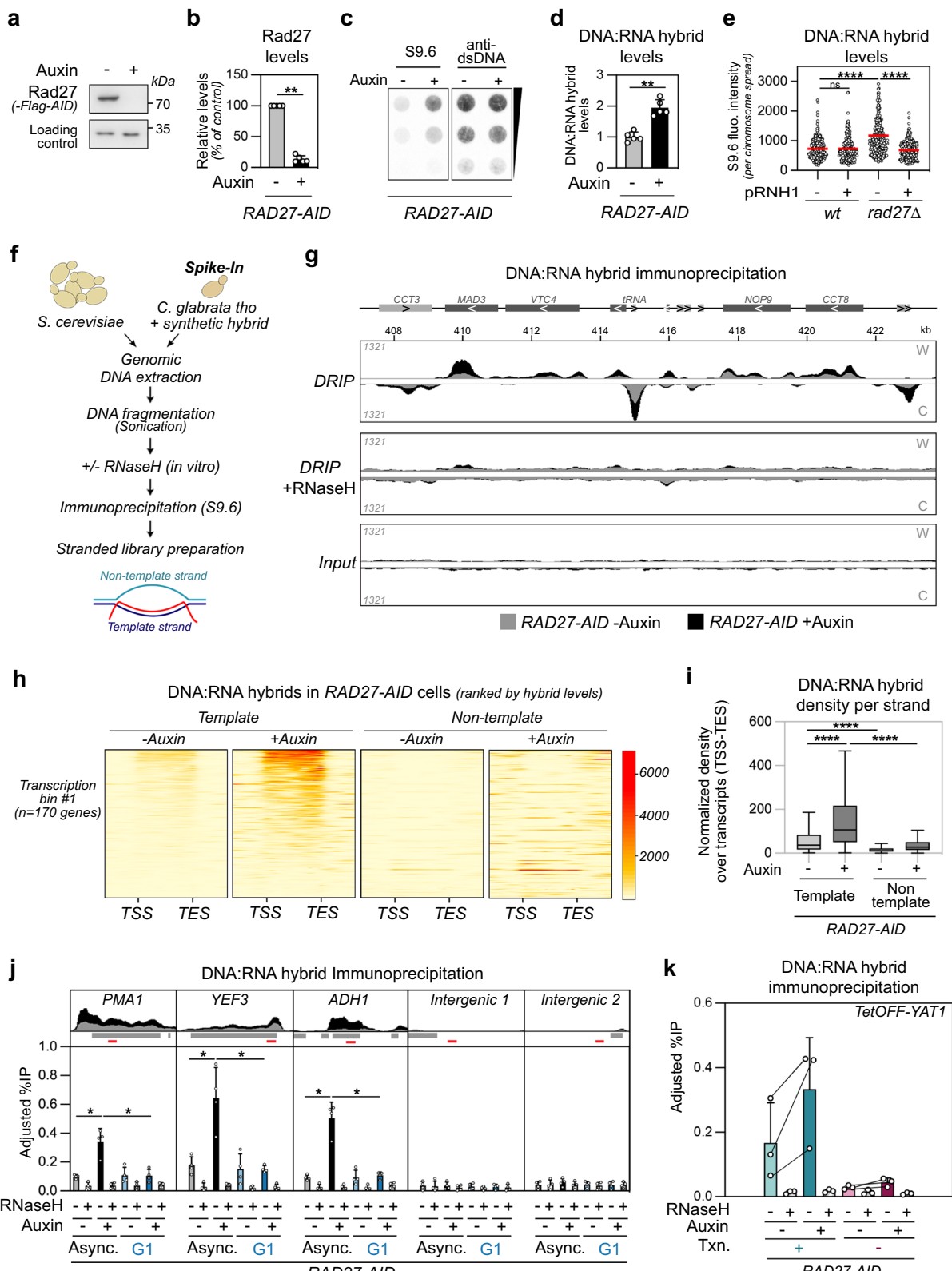

We next used different strategies to interfere with the formation of both classes of hybrids and determine their impact on genetic stability. We first prevented hybrid formation on the *YAT1* hyper-recombination reporter (Fig. 4c), either by turning off its transcription (glucose conditions), or through the insertion of an intron, which suppresses R-loop formation in *cis*[29,35]. Importantly, decreasing hybrid accumulation in both ways strongly reduced genetic instability in *tho*

mutant cells (Fig. 4d, top panel), in agreement with earlier reports[29,30,35]. In contrast, decreasing hybrid formation did not suppress the hyper-recombination phenotypes caused by the loss of Rad27 (Fig. 4d, bottom and Supplementary Fig. 6c), demonstrating that those hybrids do not detectably contribute to the associated genetic instability. To complement the assays performed with these SSA reporters, we monitored the appearance of Rad52 foci, a

**Fig. 2 | Okazaki Fragment processing defects lead to genic DNA:RNA hybrid accumulation. a** Whole-cell extracts of *RAD27-AID* cells (control or auxin-treated) were analyzed by western blot using antibodies detecting Rad27-Flag-AID (anti-Flag, top panel). Dpm1 was used as a loading control (bottom panel). **b** Rad27 levels were quantified from (**a**) (mean ± SD; *n* = 4; relative to Dpm1 and expressed as % of control; **, *p* = 0.0079). **c** DNA:RNA hybrid levels were assessed by dot blot on genomic DNAs obtained from *RAD27-AID* cells (control or auxin-treated) using S9.6 antibodies (left panel). dsDNA levels were used as loading control (right panel). Decreasing amounts of genomic DNA were loaded for quantification. **d** DNA:RNA hybrid levels were quantified from (**c**) (mean ± SD; *n* = 5; relative to dsDNA and mean control values; **, *p* = 0.0079). **e** DNA:RNA hybrid levels (*n* = 3; total number of cells: *wt*, *n* = 306, *wt pRNH1*, *n* = 280, *rad27Δ*, *n* = 306, *rad27Δ pRNH1*, *n* = 314; ****, *p* < 0.0001) were assessed by immunofluorescence using S9.6 antibodies on chromosome spreads (*pRNH1*: ectopic ScRNH1 expression). **f** Principle of strand-specific DNA:RNA hybrid immunoprecipitation (DRIP) coupled to sequencing. **g** Snapshots of strand-specific DRIP-seq coverage in *RAD27-AID* cells (gray: control; black: auxin-treated). Signals from Watson (W) or Crick (C) strands are represented. A representative replicate is featured for total genomic DNA (input), DRIP, and DRIP following in vitro treatment with RNase H.

**h** Heatmap analysis of DRIP signals at highly-transcribed genes, aligned at their Transcription Start Site (TSS) and Transcription End Site (TES), and ranked according to hybrid levels on the template strand (control conditions). Strand-specific signals are represented for both strands. 500 bp upstream the TSS and downstream the TES are displayed, and only the regions between the TSS and the TES are scaled. **i** Quantification of strand-specific DNA:RNA hybrid levels densities over transcribed regions in control or auxin-treated conditions (****, *p* < 0.0001). Box-plots are defined as above (Fig. 1j). **j** Top, Snapshots of DRIP signals at the indicated loci in *RAD27-AID* cells (gray: control; black: auxin-treated). Bottom, DNA:RNA hybrid detection (DRIP-qPCR; adjusted % of IP; mean ± SD; *n* = 4; *, *p* = 0.0286) in *RAD27-AID* cells (control or auxin-treated). Cells were either asynchronous (Async) or arrested in G1 prior to Rad27 depletion. When indicated, DNA extracts were treated with RNase H in vitro prior to immunoprecipitation. The positions of qPCR amplicons are displayed as red bars. **k** DNA:RNA hybrid detection (DRIP-qPCR; adjusted % of IP; mean ± SD; *n* = 3) in *RAD27-AID* cells (control or auxin-treated) carrying a *TetOFF-YAT1* construct. When indicated, transcription of the transgene was inhibited by doxycycline (Txn.: −). Statistical test: Two-sided Mann–Whitney–Wilcoxon rank sum test. Source data are provided as a Source Data file.

recognized proxy for global DNA damage[65]. Increased occurrence of such foci is dependent on accumulating hybrids in *tho* mutant cells, as this phenotype is reversed by in vivo over-expression of RNase H[66]. In contrast, while Rad52 foci accumulated upon Rad27 depletion, in line with our earlier reports[43,65], their formation was insensitive to RNase H-mediated hybrid removal (Fig. 4e, f). Similarly, in human cells, while FEN1 inhibition increased γH2AX DSB repair foci (Fig. 4g; Supplementary Fig. 6d, left panels), in line with previous studies[39,40], concomitant hybrid removal through RNase H1 over-expression (Supplementary Fig. 6e) did not prevent DNA damage accumulation (Fig. 4g; Supplementary Fig. 6d). Altogether, our data indicate that "post-lesion" hybrids accumulating upon OF processing defects do not contribute measurably to genetic instability.

To assess the generality of these observations, we analyzed the impact of hybrids on genetic instability in other mutants identified in our screens (Supplementary Data 1). Strikingly, when we categorized mutants ranked according to their hybrid-accumulation phenotypes (Q1–Q4; Fig. 4h), they exhibited similar levels of genetic instability with both recombination reporters (Fig. 4i and Supplementary Fig. 7a, b). Ranking the hyper-recombinant subset based on recombination indices (screen I) or frequencies (screen II) similarly revealed a lack of correlation with hybrid levels (Supplementary Fig. 7c). Thus, the global accumulation of hybrids is rarely predictive of the associated hyper-recombination rate. Next, we directly assessed the local impact of hybrids on genetic instability, by focusing on two specific Q1 mutants (Fig. 4h), which impact distinct DNA-dependent transactions, i.e., DNA replication and DNA damage checkpoint (*ctf18Δ*), or DNA repair (*rad57Δ*). We confirmed hybrid accumulation at the *YAT1* reporter locus in these mutants (Supplementary Fig. 7d) and found that their genetic instability, as measured by hyper-recombination, was independent from transcription and hybrid formation (Fig. 4j, compare plus *vs*. minus transcription). Similarly, Rad52 foci accumulation, albeit evident in both mutants, was not alleviated upon RNase H over-expression (Fig. 4k). Altogether, our data support that DNA:RNA hybrids can accumulate as a consequence of various types of DNA lesions yet do not detectably contribute to their genotoxicity.

## Discussion

Using distinctive read-outs of DNA:RNA hybrid accumulation and genetic instability (Fig. 1), we unexpectedly found that hybrids are rather a consequence than a cause of DNA lesions in multiple instances, in particular upon OF processing defects (Fig. 2). On the one hand, alleviating DNA:RNA hybrid formation through targeted transcriptional repression, intron-mediated spliceosome recruitment or RNase H over-expression, impacted neither the formation nor the fate of

recombinogenic lesions, as scored through Rad52 foci or SSA reporter analyses (Fig. 4). In contrast, stabilizing natural DNA lesions such as unprocessed flaps (Rad27 depletion) or nicks between OF termini (*CDC9* inactivation) resulted in increased hybrid levels, while rescuing flap accumulation, by over-expressing Exo1 in Rad27-depleted cells, triggered hybrid loss (Fig. 3).

Beyond OF processing defects, distinct stress or pathological situations could similarly trigger the accumulation of discontinuities such as single-strand breaks (SSB) or ssDNA gaps, and consequently lead to the secondary formation of DNA:RNA hybrids, as suspected to occur upon exposure to reactive oxygen species[67] or in the absence of post-replicative repair[9]. Importantly, our observations with other mutants assessed here (*rad57Δ*, *ctf18Δ*; Fig. 4j-k) are consistent with a model whereby ssDNA exposure precedes hybrid formation. Indeed, Rad57, a chaperone for the assembly of Rad51-ssDNA filaments[68], most likely acts at ssDNA stretches or gaps[69]. Likewise, the replication clamp loader Ctf18 activates the S phase checkpoint in response to fork stalling and the accumulation of ssDNA stretches[70]. Similar DNA discontinuities may also account for hybrid accumulation in other mutants identified in our screens, e.g., upon inactivation of the chromatin assembly factor CAF-1 (*rlf2Δ*, *cac2Δ*; Supplementary Fig. 1f). In the same line, DNA discontinuities, SSBs or gaps could also contribute to the reported accumulation of hybrids scored along with (i) replication fork pausing, as caused by dNTP deprivation upon HU treatment[26–28] or fork barrier activity[54], and (ii) replisome instability in situations of head-on (HO) transcription-replication conflicts, as triggered at dedicated reporter loci in bacteria[71], yeast[72], and human[25].

By what mechanism(s) could hybrid accumulate at flaps, SSBs, or ssDNA gaps? We do not favor a model in which such hybrids result from OF RNA primers hybridization or de novo transcription events, as proposed to occur at artificially-induced DSBs[23]. Indeed, hybrids accumulating upon defective OF processing are restricted to transcribed regions and are already detected in control cells in DRIP-seq analyses (Fig. 2). Hybrid gain is also unlikely to arise from reduced R-loop processing by Rad27, as observed in quiescent cells or at telomeres[56,73], since Rad27-E176A, which lacks direct R-loop cleavage activity in vitro, fully suppresses hybrid formation in vivo (Fig. 3). Rather, we favor two non-mutually exclusive models for hybrid formation upon defective OF processing (Supplementary Fig. 8a). On the one hand, the presence of discontinuities within DNA duplexes could promote invasion by nascent RNAs (Supplementary Fig. 8a, top panels), as proposed for in vitro or in vivo transcribed nicked templates[22,74]. In this situation, RNA invasion would not necessarily require strand exchange activities, which were shown to contribute to

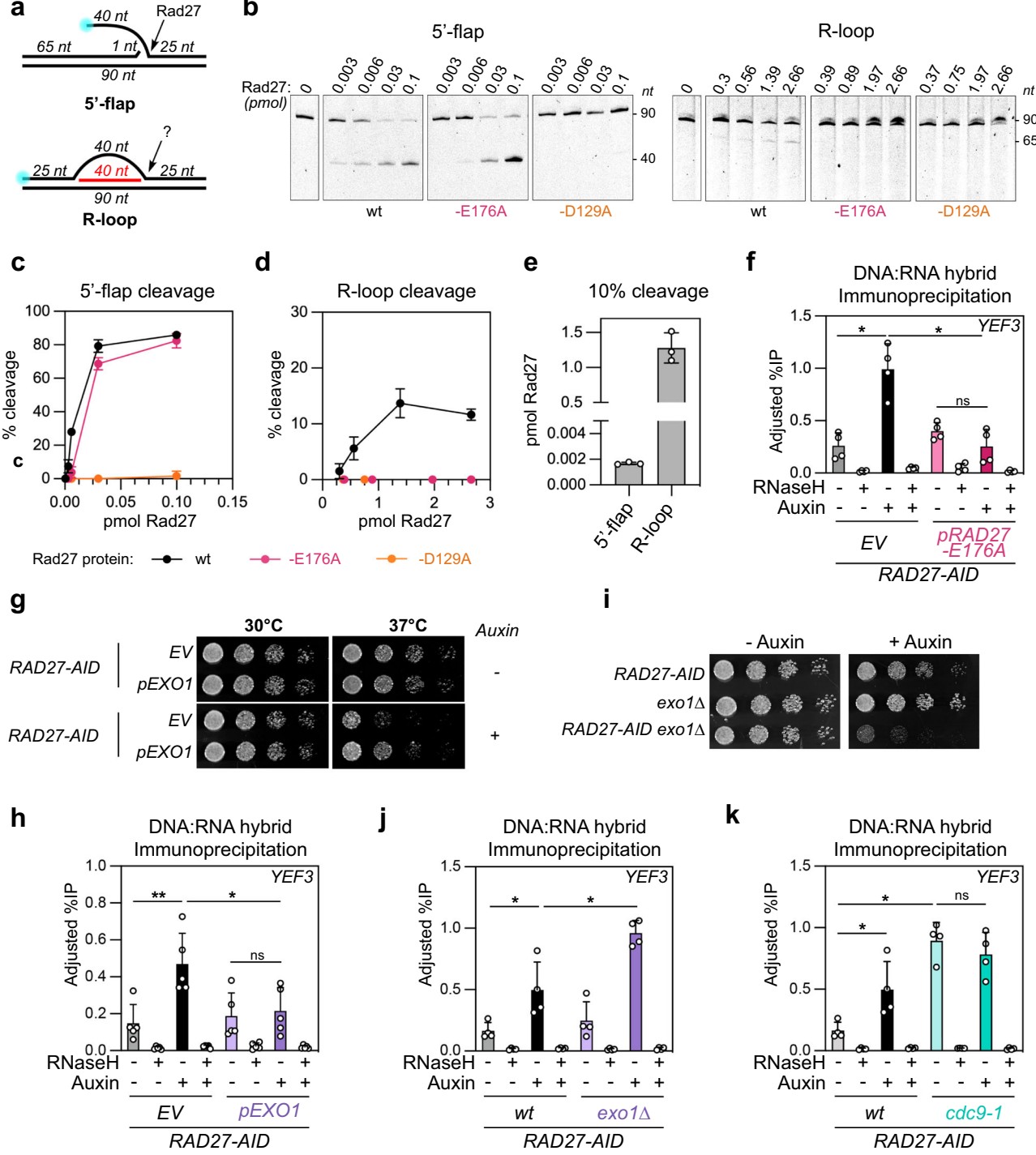

**Fig. 3 | DNA discontinuities associated with OF processing defects precede DNA:RNA hybrid accumulation. a** Schematic representation of 5′-flap and R-loop substrates used in in vitro cleavage assays. The length of DNA/RNA fragments (nucleotides, nt), the position of fluorescent labels (blue circles), and the site of Rad27 cleavage are indicated. **b** Labeled 5′-flaps (left panels) or R-loop substrates (right panels) were incubated with the indicated amounts of either *wt*, E176A or D129A purified Rad27 proteins (pmol). Cleavage products were visualized following denaturing electrophoresis. The position of uncleaved (90) and cleaved (40/65) fragments are indicated (nucleotides). **c, d** Cleavage efficiency (% of cleaved products over total labeled fragments; mean ± SD; *n* = 3) as a function of Rad27 amounts (pmol) for *wt*, E176A or D129A Rad27 variants. **e** Amounts of *wt* Rad27 proteins required to achieve 10% cleavage for 5′-flap and R-loop substrates (obtained from (**c, d**); mean ± SD; *n* = 3). **f** DNA:RNA hybrid detection (DRIP-qPCR; adjusted % of IP; mean ± SD; *n* = 4; *, *p* = 0.0286) at the *YEF3* locus in *RAD27-AID* cells (control or auxin-treated) carrying an

empty vector (*EV*) or a construct expressing the Rad27-E176A mutant protein. **g** Serial dilutions of *RAD27-AID* cells were grown at the indicated temperatures (30 °C, 37 °C) on rich medium (YPD) in the absence of the presence of auxin. Cells carried an empty vector (*EV*) or a construct over-expressing Exo1 (*pEXO1*). **h** DNA:RNA hybrid detection (DRIP-qPCR; adjusted % of IP; mean ± SD; *n* = 5; *, *p* = 0.0317; **, *p* = 0.0079) at the *YEF3* locus in *RAD27-AID* cells (control or auxin-treated) carrying an empty vector (*EV*) or a construct over-expressing Exo1. **i** Serial dilutions of the indicated strains were grown at 30 °C on rich medium (YPD) in the absence or presence of auxin. DNA:RNA hybrid detection (DRIP-qPCR; adjusted % of IP; mean ± SD; *n* = 4; *, *p* = 0.0286) at the *YEF3* locus in *RAD27-AID wt, exo1Δ* (**j**) or *cdc9-1* (**k**) derivatives (control or auxin-treated). The same *wt* control is used in (**j, k**). For all DRIP-qPCR assays, when indicated, DNA extracts were treated with RNase H in vitro prior to immunoprecipitation. Statistical test: Two-sided Mann–Whitney–Wilcoxon rank sum test. Source data are provided as a Source Data file.

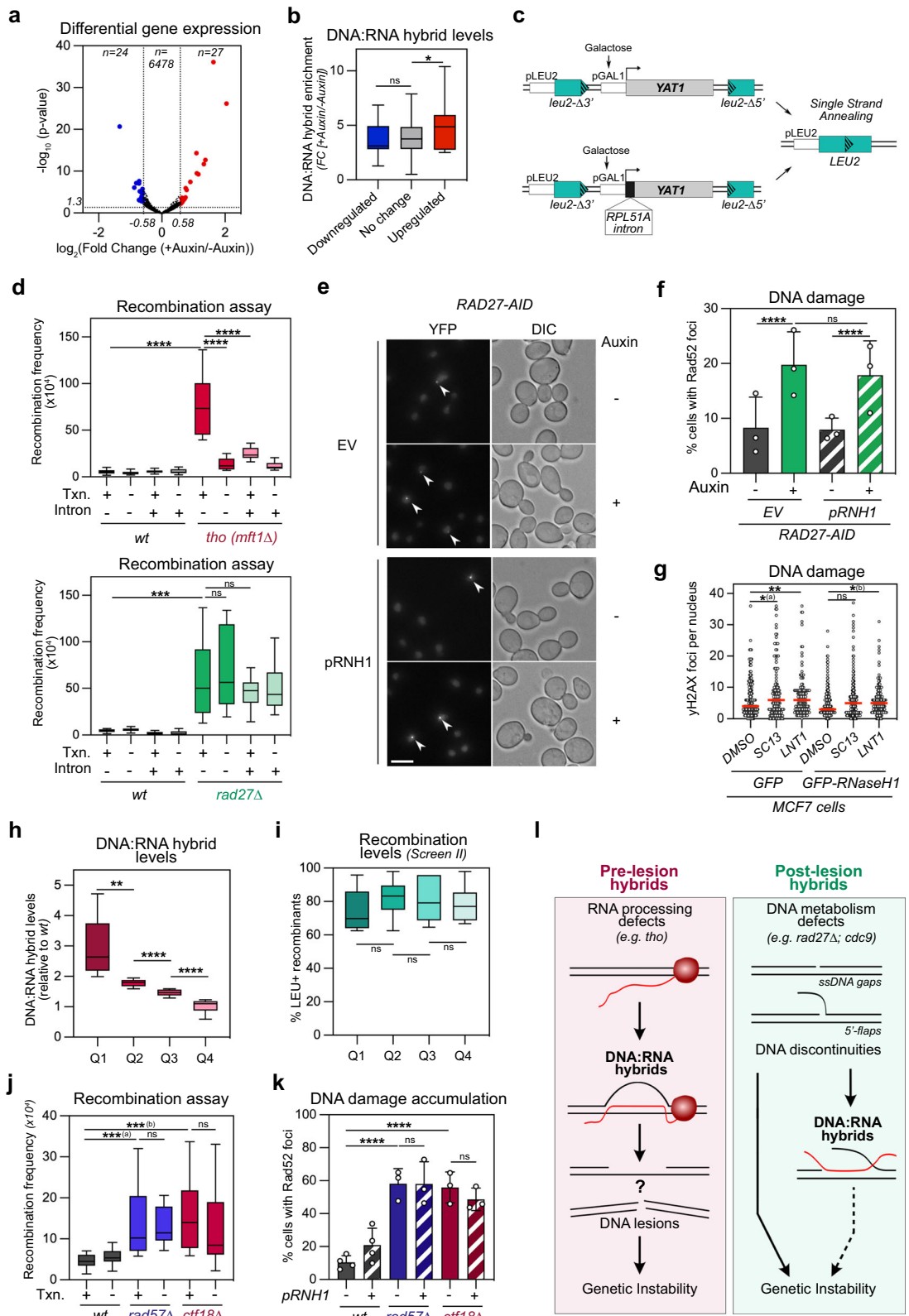

hybrid formation in vivo[75–77]. Consistently, we found that inactivation of Rad51 did not impact hybrid accumulation upon OF processing defects (Supplementary Fig. 8b). On the other hand, hybrid formation may be facilitated by changes in transcriptional dynamics (Supplementary Fig. 8a, bottom panels), as previously discussed[78] and observed in situations of RNAP II pausing[79–81]. These two mechanisms could account for hybrids forming irrespectively of the orientation of

the transcribed unit with respect to the replication fork (Supplementary Fig. 8a, compare left and right panels), as observed in our DRIP-seq dataset (Supplementary Fig. 8c). In either case, such hybrids may also transiently form during the course of normal OF processing in *wt* cells (Supplementary Fig. 4f), reminiscent of post-replicative DNA:RNA hybrids, as recently visualized behind replication forks in replicative stress models[82].

**Fig. 4 | Post-lesion DNA:RNA hybrids neither impact gene expression nor genome stability. a** RNA-seq analysis of control and auxin-treated *RAD27-AID* cells. Differential expression ($\log_2$ Fold Change [±Auxin]) and associated significance ($-\log_{10}$ *p*-value, *n* = 3) are represented for the whole transcriptome. Transcripts that are up- or down-regulated ($|\log_2 FC| > 0.58$; *p* < 0.05) upon Rad27 depletion appear in red and blue, respectively. **b** DNA:RNA hybrid enrichments following Rad27 depletion (Fold change [±Auxin]); *, *p* = 0.0411) are represented for up-regulated, down-regulated, and unaffected transcripts based on (**a**). **c** Schematic representation of inducible *GAL-YAT1* reporter constructs used in this study. **d** Recombination frequencies (fraction of *LEU*+ prototrophs; *n* = 14; ***, *p* = 0.0003; ****, *p* < 0.0001) for the indicated strains carrying *GAL-YAT1* or *GAL-intron-YAT1* transgenes repressed with glucose (Txn.: −) or induced with galactose (Txn.: +). **e** Fluorescence microscopy analysis of *RAD27-AID* cells (control or auxin-treated) expressing the Rad52-YFP fusion (*pRNH1*: ectopic ScRNH1 expression). DIC differential interference contrast. Scale bar, 5 μm. **f** Quantification of Rad52 foci from (**e**) (mean ± SD; *n* = 3; ****, *p* < 0.0001). **g** Quantification of γH2AX foci in MCF7 cells transfected with control (GFP) or RNase H1-overexpressing (GFP-RNH1) constructs. When indicated, cells were treated with SC13 or LNT1 inhibitors (*n* = 3; total number of cells counted: *GFP, DMSO n* = 149, *SC13 n* = 148, *LNT1 n* = 137; *RNH1, DMSO n* = 150, *SC13 n* = 145, *LNT1 n* = 130; *(a), p* = 0.0139; *(b), p* = 0.0471; **p* = 0.0093). **h, i** Hyper-rec mutants common to Screen I and II (*n* = 39) were split into quartiles according to DNA:RNA hybrid levels ((**h**); **, *p* = 0.0018; ****, *p* < 0.0001). Recombination levels from screen II (% *LEU*+ recombinants, (**i**)) are represented for each quartile. **j** Recombination frequencies were calculated as above (fraction of *LEU*+ prototrophs; *n* = 14; ***(a), p* = 0.007; ***(b), p* = 0.0001) for the indicated strains carrying the *GAL-YAT1* transgene and treated with glucose (Txn.: −) or galactose (Txn.: +). **k** Quantification of Rad52 foci (mean ± SD; *wt*, *n* = 4, *rad57Δ*, *ctf18Δ*, *n* = 3; ****, *p* < 0.0001) in the indicated strains (*pRNH1*: ectopic ScRNH1 expression). **l**, Model for the formation of post-lesion DNA:RNA hybrids. Box plots (**b, d, h, i, j**) are defined as above (Fig. 1j). Statistical tests: **a** Wald test corrected for multiple testing; **b, d, g–j** Two-sided Mann–Whitney–Wilcoxon rank sum test; **f, k** Two-sided Fisher exact test. Source data are provided as a Source Data file.

It is noteworthy that hybrids forming at pre-existing DNA lesions do not, as such, detectably aggravate their recombinogenicity (Fig. 4). However, it is likely that the accumulation and genotoxicity of these hybrids is restricted through alternative mechanisms, ultimately preserving genome stability and cell fitness. Consistently, we found that removing simultaneously both RNase H1 and RNase H2 activities led to strict synthetic lethality in combination with Rad27 depletion (Supplementary Fig. 8d), in line with reported systematic screens[83]. Similarly, inactivation of the Sen1 DNA:RNA hybrid helicase triggered strong growth defects in Rad27-depleted cells (Supplementary Fig. 8e).

Altogether, our observations suggest that regardless of their impact on genome stability, DNA:RNA hybrids could be functionally categorized according to their stage of formation, building upon a previous classification based only on their mapped position within gene units[16]. On the one hand, defects in nascent mRNA metabolism would trigger "pre-lesion" DNA:RNA hybrids that would result in DNA damage, a situation exemplified by THO complex or splicing mutants (Fig. 4l, left panel). On the other hand, defects in DNA metabolism associated with the accumulation of DNA discontinuities could trigger the formation of "post-lesion" hybrids (Fig. 4l, right panel), which would not necessarily impact genetic instability. Such post-lesion processes could occur to different extents depending on the type, the lifetime or the cell cycle specificity of DNA lesions, and deserve further investigation in the ever-increasing number of pathological situations associated with both DNA:RNA hybrid accumulation and genetic instability.

## Methods
### Yeast strains and growth
All *S. cerevisiae* yeast strains used in this study (listed in Supplementary Table 1) are isogenic to S288C and were obtained by homologous recombination and/or successive crosses according to standard procedures. Unless otherwise specified, cells were grown at 30 °C in standard yeast extract peptone dextrose (YPD) or synthetic complete (SC) medium supplemented with the required nutrients. To select for hygromycin-resistant cells, monosodium glutamate (0.1%) was used as nitrogen source instead of ammonium sulfate in SC media containing 300 μg/mL hygromycin B. For experiments involving thermosensitive mutants (*cdc9-1*), cells were grown at 25 °C and shifted at 30 °C for 2 h prior to analysis. For auxin-induced depletion, cells were grown in liquid SC medium containing 0.002% tryptophan and treated with 0.5 mM auxin (Indole-3-Acetic Acid) for 1 h. For transcriptional inhibition, cells were treated with 100 μg/mL 1,10-phenanthroline (Sigma) for 30 min. For synchronization, G1 arrest was triggered at 30 °C by three sequential additions of alpha-factor (2 μg/mL, 4 μg/mL then 2 μg/mL, Biotem) spaced by 40 min, and verified by microscopy observation and flow cytometry analysis. Release into the cell cycle was achieved by collecting alpha-factor arrested cells on 0.22 μm filters, washing with SC medium, and resuspending them in SC medium with or without 0.5 mM auxin for the indicated duration. For experiments involving *GAL* promoter induction, cells were grown at 30 °C in glycerol-lactate (GGL: 0.17% YNB, 0.5% ammonium sulfate, 0.05% glucose, 2% lactate and 2% glycerol) supplemented with the required nutrients prior to induction with glucose or galactose (2%) for 30 min (5 h in the case of *tho* mutants, to achieve detectable transgene induction). For experiments using TetOFF constructs, transcription was repressed by overnight growth in the presence of 5 μg/mL doxycycline (Sigma). Growth assays were performed by spotting serial dilutions of exponentially-growing cells on solid medium and incubating the plates at the indicated temperatures for 2 days. When indicated, YPD plates were supplemented with 0.5 mM auxin or 0.01% MMS.

### Cell lines and cell culture
Parental MCF7 and MCF7-RHINO cell lines were grown as monolayers in Minimum Essential Medium (MEM) with GlutaMax supplemented with 10% (v/v) fetal bovine serum, 1% (v/v) 100X MEM non-essential aminoacids and 1% (v/v) 100 mM sodium pyruvate. All cells were maintained at 37 °C in a humidified atmosphere with 5% $CO_2$.

The MCF7-RHINO cell line was made by co-transfecting MCF7 cells with the plasmids MK232-TRE3GS-RHINO-TetON-Puro plasmid and eSpCas9(1.1)_No_FLAG_AAVS1_T2 (a gift from Yannick Doyon, Addgene plasmid #79888[84]) at a ratio of 8:1. The transfected MCF7 cells were re-plated 24 h after transfections and selected with Puromycin (1 μg/mL). The MCF7-RHINO cell population was then induced with 1 μg/mL doxycycline for 18 h before sorting cells expressing mNeonGreen fluorescence (RHINO). For live cell experiments, cells were seeded into 35-mm Petri dishes with 14-mm glass-bottom microwell (coverglass thickness no. 1.5, MatTek Corporation). MCF7-RHINO cells, incubated with 1 μg/mL doxycyclin for 6 h before drug treatment, and RHINO-transfected MCF7 cells were treated with SC13 (MedChemExpress) or LNT1 (Tocris Biotechne) at a final concentration of 50 μM (or an equivalent volume of DMSO as control) for 12–24 h before microscopic observation and image acquisition. For RHINO mutant experiments, MCF7 cells were transiently transfected with 500 ng of RHINO W43A or KK59/60AA constructs[41] using Lipofectamine 3000 reagent (Thermo Fisher Scientific), according to the manufacturer's protocol. For RNase H1 over-expression experiments, MCF7 cells were seeded on glass coverslips no. 1 (Normax) in 6-well plates and transfected with 1 μg of either pUBC-GFP-RNase H1 construct or the empty vector (pUBC-GFP-C1) as above. After 6 h of incubation, cells were treated with SC13 or LNT1 at a final concentration of 50 μM (or an equivalent volume of DMSO as control) for 24 h prior to immunofluorescence. Hybrid removal by RNase H1 was checked by transfecting MCF7-RHINO cells with mScarlet-i-tagged human RNase H1 constructs, either wildtype (wt) or catalytically dead (D210N).

## Plasmids

The construction of the plasmids used in this study (listed in Supplementary Table 2) was performed using standard PCR-based cloning techniques and checked by sequencing.

## Systematic hyper-recombination screens

Systematic mating and pinning of libraries were achieved using a robotized device (RoToR, Singer Instruments), following the manufacturer's instructions. The cytometry-based hyper-recombination screen (screen I, Fig. 1a) was performed with the Yeast Knockout mat alpha collection (BY4742 and BY4739 background), re-arrayed together with parental *wt* into 96-well plates organized according to chromosome position. A mating-based procedure, systematic hybrid loss of heterozygosity[33], was used to simultaneously introduce the *LYS2-YFP* reporter plasmid in 4831 arrayed deletants. Briefly, 16 donor strains, each containing a conditional centromere on one of the 16 yeast chromosomes and further transformed with the reporter plasmid, were mated to arrays of deletants in the corresponding chromosomes. Selectable loss of the conditional chromosome and endoduplication of the chromosome carrying the deletion generate 2n hybrid diploids homozygous for the gene deletion (Supplementary Fig. 1a) and complemented for spurious recessive mutations that are typical of mutant libraries[85] and could influence screen outcome[33]. During these steps, cells were grown in the absence of lysine to ensure the maintenance of the reporter plasmid under its non-recombined form (*LYS+*). Strains were then transferred to 96-well liquid culture plates and grown at 23 °C for 2 days to allow recombination, YFP expression, and culture saturation prior to flow cytometry. 100,000 cells were measured for every sample by flow cytometry with Cell-Quest software using a BD FacSCAN or BD FACSCalibur cytometer. Cell gating and counting were achieved using OMIQ (Dotmatics) using as references control non-fluorescent strains containing empty vector plasmids. To assess recombination, the fraction of *YFP+* cells was counted for each sample and normalized to the median value obtained for the corresponding plate. 261 strains with measurements above the 95th percentile threshold for at least two replicates out of four were selected for further verification and re-tested at least eight times, simultaneously with *wt*. To rank every mutant strain according to the likelihood that it differed from control, all *wt* sample measurements (*n* = 4642) were combined to create ten statistical comparison groups. Every individual strain measurement was then normalized to its respective comparison group by determining the proportion (%) of that *wt* distribution which was lower in magnitude, thereby yielding recombination indices between 0 and 100 (Supplementary Data 1). A value of 100 means that all *wt* measurements were lower than the mutant measurement, implying hyper-recombination, while a value of 0 indicates hypo-recombination. Normalized values were compared to the pooled *wt* group to determine a Mann–Whitney *p*-value (Supplementary Data 1). To determine a cutoff point for downstream analyses, all reported genetic and physical interactions between ranked mutants (as in the Biogrid release 4.4.226 from 2023/09/25) were plotted in a 2-dimensional graph, using the CLIK (cutoff linked to interaction knowledge) software[34], thus defining a sub-library of 114 highly-interacting, hyper-recombinant mutants.

Mutants in this subset were re-isolated, verified by PCR for the deletion of interest, and re-arrayed together with parental *wt* and recombinant-deficient *rad52Δ* strains into 1536 plates (each mutant strain in 16 replicates). For the high-throughput replica pinning hyper-recombination screen (screen II, Fig. 1a), the sub-library was mated to an universal donor strain containing conditional centromeres on every chromosome and transformed with the *YAT1-LEU2* reporter plasmid, and deletant haploid strains carrying the construct were further selected as previously described[86]. During these steps, cells were grown in the presence of hygromycin B to ensure the maintenance of the reporter plasmid under its non-recombined form (*hygR*). To

achieve expression of the *GAL-YAT1* transgene, cells were further transferred on SC medium containing 2% raffinose as carbon source, prior to two successive replica pinning onto SC minus leucin containing 2% galactose medium to select for recombinants. Each plate was pinned onto three selective plates, yielding 48 replicates per mutant strains in total. Following scanning, colonies were assessed as positive (*LEU+*) as previously described[87], yielding recombinant frequencies, and statistical difference with *wt* (3264 replicates) was determined using a two-sided Fisher exact test (Supplementary Data 1). Strains poorly or not growing on raffinose or galactose media (*n* = 37) were not further considered in downstream analyses to avoid selection bias. Since *hpr1Δ* THO complex mutants exhibited decreased fitness in these pinning steps, we used instead another mutant of the same complex, *mft1Δ*, with similar reported phenotypes yet milder growth defects[88]. Gene Ontology analysis was performed using the Gene Ontology Slim Term Mapper (www.yeastgenome.org/goSlimMapper). Conservation analysis was performed using the DRSC Integrative Ortholog Prediction tool (www.flyrnai.org/cgi-bin/DRSC_orthologs.pl).

## DNA:RNA hybrid quantification methods

Dot-blot detection of DNA:RNA hybrids was performed as previously described[29]. Briefly, total genomic DNA was phenol-extracted and isolated by ethanol precipitation. Decreasing amounts of DNA preps were adsorbed onto nylon membranes (Hybond-N+, GE Healthcare) which were incubated with the following mouse antibodies: anti DNA:RNA hybrids (S9.6, Kerafast; 0.3 μg/mL in Tris-Buffered Saline [TBS], 0.5% Tween-20, 5% skimmed milk); anti-double-stranded DNA (HYB331-01, Santa Cruz Biotechnology; 1:5000 in TBS, 0.5% Tween-20, 5% BSA). Following incubation with anti-mouse peroxidase-conjugated antibodies (Jackson ImmunoResearch), membranes were revealed using chemiluminescent reagents (Supersignal, Thermo Fisher Scientific) and imaged with a ChemiDoc MP Imaging System (Bio-Rad). Relative DNA:RNA hybrid and total DNA amounts were determined using serial dilutions of a reference sample as a standard. Specificity of the dot-blot signals was confirmed by treating the DNA extracts with RNase H (New England Biolabs) prior to blotting[29].

DNA:RNA hybrid IP coupled to sequencing (DRIP-seq) was performed according to a published procedure[52], with the following modifications. One hundred OD of exponentially growing yeast cells were collected, mixed with a synthetic DNA:RNA hybrid spike-in[52] (Supplementary Table 3), in a 1:1 ratio with cell count, and *hpr1Δ C. glabrata* cells, in a 95:5 ratio, prior to lysis. Genomic DNA was isolated as above, and DNA concentration was estimated using the Qubit HS DNA kit. Purified DNA was diluted to 275 ng/μL and fragmented in a Covaris focused-ultrasonicator (Covaris M220) with the following parameters: Peak Incident Power = 50 W, Duty Factor = 20%, Cycles per Burst = 200, 7 cycles, 10 s sonication, 20 s rest, 7 °C. Seventy-five micrograms of DNA were subsequently used for IP, diluted in 450 μL with Fast Digest buffer (Thermo Fisher Scientific), and incubated for 30 min at 37 °C, in the presence or absence of 350 units of RNase H (New England Biolabs). An aliquot of each reaction was taken as an input fraction (5%); the remaining sample was diluted fourfold with FA1 buffer (0.1% SDS, 1% Triton X-100, 10 mM HEPES pH 7.5, 0.1% sodium deoxycholate, 275 mM NaCl), and mixed 1 h at 4 °C in the presence of 5.25 μg of S9.6 purified antibody. IP mixtures were further mixed with Dynabeads Protein G (Thermo Fisher Scientific) for 2 h at 4 °C. Beads were then washed as follows: once with FA1 buffer, once with FA1 buffer containing 360 mM NaCl; twice with 10 mM Tris pH 8, 250 mM LiCl, 0.5% Nonidet-P40, 0.5% deoxycholate, 1 mM EDTA and once with 10 mM Tris–HCl pH 8, 1 mM EDTA. Elution was achieved through a 20 min incubation at 65 °C in the presence of 50 mM Tris pH 8, 10 mM EDTA, 1% SDS, followed by deproteinization with 0.16 μg/μL proteinase K in the presence of 250 mM NaCl for 1 h at 42 °C and 30 min at 65 °C. Input and immunoprecipitated DNAs were purified with the QIAquick

DNA purification kit (Qiagen) according to the manufacturer's instructions. Libraries were created using the xGen ssDNA & Low-Input DNA Library Prep Kit (Integrated DNA Technologies) according to the manufacturer instructions, with 16 cycles of amplification, and further quantified by Qubit and Tapestation (Agilent). Paired-end sequencing was performed on an Illumina Novaseq platform (Novogene). Reads were trimmed from 3′ and 5′ ends until the final base had a quality score >30 using Cutadapt 3.4, discarding reads left with <15 bp, and further aligned to the SacCer3 (*S. cerevisiae*) and CBS138 (*C. glabrata*) genomes using Bowtie2 (version 2.5.0), with the following parameters: $D = 20$, $R = 3$, $N = 0$, $L = 20$, $I = S,1,0.5$, $X = 500$. Aligned reads were filtered to retain only the reads mapped in a proper pair with a quality score >30 using Samtools (version 1.8), and duplicates were removed using Rmdup (version 2.0.1). For *S. cerevisiae*, aligned reads were split based on their strand origin using BamCoverage (version 3.5.4), with a bin size of 20 bp, no scaling, and only including reads originating from fragments from the Watson or Crick strands. Densities were calibrated using an occupancy ratio defined for each condition based on the number of *C. glabrata* reads in input and immunoprecipitated samples, as previously described for calibrated ChIP-seq[53]. Coverage was visualized using Integrative Genomics Viewer. Previously described transcript annotations[89] and transcription frequencies[90] were used to compute DNA:RNA hybrid densities per transcribed regions. Correlation analyses were performed with Prism v10.1.1. To define the co-directional (CD) or head-on (HO) orientations for each gene, we used previously published OK-seq data[91]. The direction of replication for each gene was determined by calculating the slope of the average OK-seq signal dataset (GSM999266 [https://www.ncbi.nlm.nih.gov/geo/query/acc.cgi?acc=GSM999266]; GSM999266_average_A.tab file). This replication direction was then cross-referenced with the transcriptional orientation of each gene to classify it as either CD or HO relative to the replication fork.

DRIP-qPCR was performed as previously described[29,35]. Genomic DNA was isolated from 25 OD of exponentially growing yeast cells mixed with the synthetic DNA:RNA hybrid spike-in (1:1 ratio with cell count) and quantified with a Nanodrop. 40 µg of nucleic acids were further digested by a cocktail of restriction enzymes (EcoRI, HindIII, XbaI, SspI, BsrGI; FastDigest enzymes; Thermo Fisher Scientific) for 30 min at 37 °C in a total volume of 100 µL. Specificity of the DRIP signal was determined by including 20 units of RNase H (New England Biolabs) in the digestion reaction. An aliquot of the digested DNA was taken as an input fraction (5%); the remaining sample was diluted fourfold with FA1 buffer and mixed overnight at 4 °C with 0.3 µg of S9.6 purified antibody. IP mixtures were further mixed with protein G Sepharose beads (GE Healthcare) for 1 h at 4 °C. Beads were then washed and DNA was purified as described above. Input and immunoprecipitated DNAs were further quantified by real-time PCR with a LightCycler 480 system (Roche) using SYBR Green incorporation according to the manufacturer's instructions. Percentages of IP were normalized to the percentages of IP obtained for the synthetic spike-in, resulting in adjusted percentages (%) of IP.

### Gene expression analyses

Total RNAs were purified from 10 OD of cells using the Nucleospin RNAII kit (Macherey Nagel) according to the manufacturer's instructions. For quantitative PCR (RT-qPCR), RNAs were reverse-transcribed using random hexamers (P(dN)6, Roche) and Superscript II reverse transcriptase (Thermo Fisher Scientific), and cDNAs were further quantified by real-time PCR as above. For transcriptome analysis, directional libraries were prepared with rRNA removal, and paired-end sequencing was performed on an Illumina Novaseq 6000 platform (Novogene). Reads were trimmed for adapter sequence from 3′ end, and from both 3′ and 5′ end until the final base had a quality score >30 using Cutadapt version 3.4, discarding reads left with <15 bp. Trimmed reads were aligned to the SacCer3 genome using RNA STAR version

2.5.0, using the following parameters: sjdbOverhang=99, quantMode=Per gene read count, ENCODE options for advanced options of the mapping stage and SGD (www.yeastgenome.org) gene annotations. Aligned reads were filtered to retain only uniquely mapped reads using bamTools version 2.5.2. Reads were then scaled to a scaling factor defined as 1/millions of uniquely mapped reads for each sample using bedtools GenomeCoverage version 2.30.0. Finally, differential expression analysis was performed using DESeq2 version 2.11.40.8.

RNA polymerase II distribution was determined by ChIP using anti-Pol II largest subunit antibodies (anti-Rpb1, BioLegend) as previously reported[92]. Input and immunoprecipitated DNA amounts were quantified by real-time PCR.

### Expression of recombinant proteins and in vitro cleavage assays

His-tagged Rad27 variants (*wt*, D179A, and E176A) were expressed in Rosetta (DE3) *E. coli* cells transformed with the corresponding plasmids and grown in LB medium supplemented with ampicillin. Expression of the recombinant proteins was induced with 0.5 mM isopropyl-β-D-thiogalactopyranoside (IPTG) at 16 °C for 16 h, following cold and chemical shocks (4 °C, 2% ethanol). Bacterial pellets were collected by centrifugation and frozen in liquid nitrogen. Pellets (1 g) were resuspended in His lysis buffer (20 mM $Na_2HPO_4$ pH 7.5, 500 mM NaCl, 10 mM imidazole, 0.2%(v/v) Triton X-100, 1 mM $MgCl_2$, 4 µg/mL pepstatin A, 180 µg/mL PMSF), treated with 0.5 mg/mL lysozyme (Thermo Fisher Scientific) for 1 h at 4 °C and lysed by sonication. Lysates were treated for 15 min at 4 °C with 0.5% Sarkosyl, supplemented with 0.8% Triton X-100 and centrifuged at $10,000 \times g$ for 20 min at 4 °C. Supernatants were further incubated with Ni-NTA agarose (Qiagen) for 2 h at 4 °C. Beads were then washed four times with His lysis buffer and eluted four times with the same buffer containing 500 mM imidazole and 1% Triton X-100. Following purification, eluate fractions were pooled, dialyzed overnight at 4 °C against 20 mM Hepes KOH pH 7.9, 0.1 M KCl, 0.1 mM DTT, and supplemented with 10% glycerol before storage at −80 °C.

R-loop and 5′-flap synthetic substrates were prepared using DNA or RNA oligonucleotides (listed in Supplementary Table 3), some of which were labeled with fluorescein (FLO) or Cy5 fluorophores at their 5′ terminus. R-loop substrates were previously described[93] and flap substrates were designed to have the same nucleotide content and sequence at the cleavage site. Substrates were prepared by heating oligonucleotide mixtures at 95 °C for 5 min and gradual cooling (1 °C per 2 min) to room temperature. The following oligonucleotides were used at the indicated concentrations: 5′-flaps, common-90-R (5 mM), [FLO]-Flap-65-F (2.5 mM), Flap-66-F (10 mM); R-loops, common-90-R (5 mM), [Cy5]-R-loop-RNA-F (12.5 mM), [FLO]-R-loop-90-F (2.5 mM). Substrates were purified by polyacrylamide gel electrophoresis in 1X TBE in a neutral 10% gel, eluted from gel slices in 10 mM Tris pH8 150 mM NaCl, aliquoted, and stored at −20 °C.

Cleavage reactions (15 µL) containing 13 nM labeled substrate and 0.003–2.66 pmol of purified Rad27 variants in 50 mM Tris pH 8, 50 mM NaCl, 5 mM $MgCl_2$, 0.6 mM DTT were incubated for 20 min at 37 °C. Reactions were stopped by adding 6 µL of 30% glycerol, 50 mM EDTA. Cleavage products were separated by polyacrylamide gel electrophoresis in 0.5X TBE in a denaturing 15% gel containing 7.9 M urea, visualized with a ChemiDoc MP Imaging System (Bio-Rad) and quantified using ImageJ (National Institutes of Health).

### Protein extraction and Western blot analysis

Yeast whole-cell extracts were obtained by the NaOH-TCA procedure[94]. Purified recombinant proteins were resuspended in Laemmli sample buffer prior to electrophoresis. Protein samples were separated on 4–12% precast SDS-PAGE gels (Thermo Fisher Scientific). Proteins were either detected by Coomassie blue staining or by western-blot following transfer to PVDF membranes. The following validated antibodies were used: anti-Flag monoclonal antibody (M2, Sigma), 1:2000;

anti-Dpm1 monoclonal antibody (Thermo Fisher Scientific), 1:1000. Membranes were incubated with anti-mouse peroxidase-conjugated antibodies and revealed as above.

## Hyper-recombination assays

Independent clones transformed with direct repeat reporters (*YAT1-LEU2*) were individually resuspended in 1 mL GGL medium, grown for at least 2 h at 30 °C and induced with glucose or galactose (2%) for either 5 h (Fig. 4d, top panel) or 30 min (Fig. 4d, bottom panel and Fig. 4j). Cells were further resuspended in water and appropriate dilutions were plated on SC medium lacking leucine to estimate the number of *LEU+* recombinants, or complete medium to estimate cell survival. Plates were incubated for 2 days at 30 °C prior to colony counting. Hyper-recombination rates were defined as the proportion of *LEU+* prototrophs. Since auxin can impair long-term growth in SC media[95], hyper-recombination assays were performed using *rad27Δ* instead of *RAD27-AID* strains. For each genotype, 14 clones from at least 3 independent experiments were analyzed.

## Cell imaging

Exponentially-growing cells expressing the Rad52-YFP fusion were harvested by centrifugation and mounted on slides for imaging. Live cell images were acquired using a DM6000B Leica microscope with a 100×, NA 1.4 (HCX Plan-Apo) oil immersion objective, a CCD camera (CoolSNAP HQ; Photometrics), and a piezo-electric motor (LVDT; Physik Instrument) mounted underneath the objective lens. Z-stack sections of 0.2 μm were acquired, scaled equivalently, and 3D-projected using MetaMorph 7.8.13.0 (Molecular Devices). For each strain and replicate, at least 25 cells were counted and statistical analyses were performed by comparing the proportion of Rad52 foci-containing cells in between conditions.

Yeast cells were prepared and fixed for chromosome spreads and S9.6 staining as described[50]. In brief, spreads were probed with a 1:1000 dilution of S9.6 antibodies (Kerafast, #ENH001), followed by 1:1000 secondary Alexa Fluor 568 goat anti-mouse antibodies (Invitrogen). Chromosome spreads were imaged using an Objective HCX PL APO 1.40 NA oil immersion 100× objective on an inverted Leica Dmi8 microscope with appropriate filter sets. The images were captured at room temperature using a scientific complementary metal oxide semiconductor (sCMOS) camera (ORCA Flash 4.0 V2; Hamamatsu), collected, and analyzed using ImageJ. The fluorescence intensity of individual stained nuclei was measured, and pooled data from >250 cells per condition (corresponding to three independent experiments) was plotted.

R-loop imaging in human cell lines was achieved by expressing the RHINO reporter, which consists of a tandem array of three hybrid-binding domains from human RNase H1 fused to the mNeonGreen fluorescent protein. Live cell imaging was performed on a 3i Marianas SDC spinning disk confocal imaging system (Intelligent Imaging Innovations Inc.) using a microscopy setup previously described[41]. The MCF7-RHINO cells were imaged with z-stacks covering the complete volume of cell nuclei encompassing 30–40 optical slices at an optimal z-slice interval of 0.27–0.32 μm using the 100× objective. Microscopy images were deconvoluted using the Iterative Deconvolve 3D plugin in the FIJI software package. Standard settings with three iterations were applied. The point spread function was generated using Diffraction PSF 3D plugin in FIJI by applying the microscope specification and settings for image acquisition. Image stacks were analyzed in the ICY image analysis package[96] as described before[41].

For γH2AX immunofluorescence, cells were briefly washed with 1X PBS and fixed with 3.7% Formaldehyde in 1X PBS for 10 min, followed by two washing steps with 1X PBS. Next, cells were permeabilized with 0.25% Triton X-100 in 1X PBS for 10 min, washed twice with 1X PBS and incubated in blocking solution (0.2% fish skin gelatin solution in PBS) for 30 min. Incubation with primary mouse anti−γH2AX antibodies

(Millipore; 1:200 dilution in blocking solution) was performed for 1 h at room temperature in a humidified chamber. Next, cells were washed three times with 0.05% Tween-20 in 1X PBS for 5 min each, and incubated with secondary goat anti-mouse antibodies conjugated with Alexa Fluor 647 (Thermo Fisher Scientific; 1:200 dilution in blocking solution) for 1 h at room temperature in a humidified chamber. Cells were then washed three times, 5 min each, with 0.05% Tween-20 in 1X PBS, incubated with 1 μg/mL DAPI in 1X PBS for 4 min at room temperature, and washed again as above. Finally, the coverslips were embedded in Fluoromount G mounting medium (Thermo Fisher Scientific) on microscopy glass slides and imaged with the same setup as described above for RHINO cells. The γH2AX foci number quantification was performed in ImageJ by a custom-written macro which segments cell nuclei according to the DAPI channel signal, followed by counting local maxima in a maximum intensity projection of the γH2AX staining channel image stack.

## Quantification and statistical analysis

The experiments were not randomized and the investigators were not blinded to allocation during experiments and outcome assessment. (*n*) values were chosen in accordance with standard practices and correspond to the number of biological replicates (e.g., independent cultures). Error bars correspond to standard deviations. The following statistical tests were used to evaluate the statistical differences between strains/conditions or proportions and computed with Prism v10.1.1 or R v4.3.1: Two-sided Mann–Whitney–Wilcoxon rank sum test; Two-sided Fisher-exact test; Hypergeometric test; Wald test corrected for multiple testing using the Benjamini and Hochberg method. Box plots were represented according to Tukey's definition using Prism v10.1.1. Outliers were not represented in Figs. 2e, 2i, 4b, 4d, 4g, 4j and Supplementary Fig. 3c for clarity but have been included in statistical analyses. The following convention was used for reporting statistical significance: *$p < 0.05$; **$p < 0.01$; ***$p < 0.001$; ****$p < 0.0001$; ns, not significant.

## Reporting summary

Further information on research design is available in the Nature Portfolio Reporting Summary linked to this article.

## Data availability

The complete NGS data generated during this study are available in NCBI's Gene Expression Omnibus under GEO series accession numbers GSE265894 (DRIP-seq) and GSE265892 (RNA-seq). All the other data supporting the findings of this study are available within the paper and its Supplementary Information. Source data are provided with this paper.

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

## Acknowledgments

We are very grateful to Masato Kanemaki, Doug Koshland, Yannick Doyon, Xiang Dong Fu, and Domenico Libri for reagents; to Nicolas Valentin (Imagoseine) and Patricia Woni for valuable assistance with flow cytometry and media preparation, respectively; to the Bioimaging Unit of Instituto de Medicina Molecular—João Lobo Antunes for technical support; to Marine Guillay, Lounis Yakhou, Adam Taheraly and Franco Magliocchetti for technical help with systematic screens; to other members of the Palancade lab for fruitful discussion; and to Vincent Vanoosthuyse, Anna Babour and Domenico Libri for their critical reading of the manuscript. This work was supported by Agence Nationale pour la Recherche (ANR-18-CE12-0003 and ANR-21-CE12-0040, to B.P.), Fondation ARC pour la recherche contre le Cancer (projet ARC PJA20181208112, to B.P.), Ligue Nationale contre le Cancer (comité de Paris RS24/75-59, to B.P.), the NIH (GM50237, to R.R.), the BioSPC PhD program (to Ra.M.M., M.Z., and A.P.), the Fondation Pour la Recherche Médicale (FDT202304016590, to Ra.M.M. and AAP Equipe FRM 2020 EQU202003010245, to P.H.G.) and the "Who am I?" laboratory of excellence (grant numbers ANR-11-LABX-0071 and ANR-18-IDEX-0001, fellowship to Ra.M.M).

## Author contributions

Conceptualization, Ra.M.M., R.R., B.P.; Methodology, Ra.M.M., S.P., R.M.M., R.J.D.R., P.H.G., P.C.S., S.F.A., R.R., B.P.; Investigation, Ra.M.M., S.P., M.Z., R.M.M., C.G., A.K., S.S., C.S.M., A.P., A.L., R.J.D.R., O.L., B.P.; Formal analysis, Ra.M.M., S.P., M.Z., R.J.D.R., B.P.; Writing—original draft, Ra.M.M., R.R., B.P.; Writing—review and editing, all authors; Visualization, Ra.M.M.; Supervision, P.H.G., P.C.S., S.F.A., R.R., B.P.; Project administration, B.P.; Funding acquisition: B.P., R.R., Ra.M.M., M.Z., A.P., P.H.G.

## Competing interests

The authors declare no competing interests.
