## [Transparent Peer Review file · Nature Communications]

DNA lesions can frequently precede DNA:RNA hybrid accumulation

Corresponding Author: Dr Benoit PALANCADE

Version 0:

Reviewer comments:

Reviewer #1

(Remarks to the Author)

Mangione et al. report on DNA:RNA hybrids forming at DNA replication-associated discontinuities of the DNA double-helix, such as unprocessed DNA flap structures and unligated Okazaki fragment. They conclude that DNA:RNA hybrids do not only form prior to DNA lesions, causing genome instability (“pre-lesion” hybrids), but commonly also as consequence of DNA lesions (“post-lesion” hybrids), without necessarily enhancing genetic instability. This is an important finding, given that DNA:RNA hybrids and especially R-loops have been investigated primarily as DNA damage-inducing structures. The genetic screens and validation experiments seem overall well conducted and the paper is very well written. If “post-lesion” hybrids are indeed a frequent and common consequence of DNA discontinuities, the manuscript provides an important counterpoint to the prevailing view on R-loop-induced genotoxicity.

Main points:

- 1) The manuscript focusses primarily on yeast, but the results do suggest that human FEN1 has a similar role as Rad27 in suppressing flap-induced hybrids. However, only a single experiment with FEN1 is shown with limited specificity controls. In light of the broad interest in DNA:RNA hybrids and their role in human disease, it would be important to strengthen this point and extend the analysis of FEN1 inhibition. Specifically:
 - a. Include specificity controls for RHINO-mediated detection of hybrids (e.g. a binding-deficient mutant, as done by Martin et al., in reference 40), provide representative images, and validate FEN1 inhibition.
 - b. Include an orthogonal approach (e.g. DRIP-qPCR/seq) and test if DNA discontinuities upon FEN1 inactivation precede and are sufficient for hybrid formation.
 - c. Include a second FEN1 or FEN1/EXO1 inhibitor (e.g. LNT1).
 - d. Test if post-lesion hybrids in mammalian cells impact genome stability.

2) While the conclusion that hybrids can form as “post-lesion” hybrids seems overall well supported by the data, the functional implications are less clear. The authors found no evidence for enhanced recombination frequency or enhanced DNA damage measured by Rad52 foci, but it is hard to believe that such hybrids would be completely neutral for genome stability. Would such hybrids become genotoxic and lead to genetic instability in certain contexts? The authors should try to identify conditions to which “post-lesion” hybrids sensitize (either by a hypothesis-driven approach or genetic screens).

Additional points:

- 1) The authors conclude that they are scoring mostly transcription-dependent hybrids (Fig. 2). In the DRIP-qPCR experiments, this could probably be tested directly by adding a transcription inhibitor to the + auxin conditions.
- 2) Examples from previous studies, where hybrids or R-loops were already observed as consequence of replication stress (references 25-28), could be discussed in the light of the new findings from the current study in more depth in the discussion section of the manuscript.

Reviewer #2

(Remarks to the Author)

In this manuscript, the authors bring evidence that DNA damage triggers the formation of genic DNA-RNA hybrids. Yeast is the model organism. The authors first looked for genes whose deletion is responsible for both recombination events and R-loop accumulation. Several factors were found but the authors particularly highlight 2 genes involved in Okazaki fragment processing (*rad27* (human FEN1) and *cdc9*). The authors claim that the R-loop accumulation in these mutants is not directly responsible for the recombinogenic events because these would mostly be occurring due to defective OF processing. Using these screens as the premise for the study, they rather hypothesize that the hybrid accumulation is triggered by the lesions occurring following OF processing defects.

They next focus on Rad27 for the study of the R-loop formation timing using AID system.

The article is clear and well-written. The study is well conducted with a mix of cell work and biochemical approaches, and the evidence brought by the authors is convincing.

I have only a few suggestions and questions.

I would suggest illustrating graph 1J with representative images of the hybrid accumulation using the RHINO probes.

In Figure 2, the authors show the replication-dependency of hybrid formation by synchronizing the cells in G1. Replication can trigger DNA damage in other ways than by defective OF processing. Can Emetine which prevents OF formation be used to strengthen the conclusion that hybrid formation is specifically triggered by OF processing defects?

The authors show that hybrid accumulation depends on transcription and occurs because of flap accumulation. However, they also show that transcription is not increased upon *rad27* deletion which could be counter-intuitive. The authors discuss this in the discussion section but do not mention a possible formation in trans. Can these hybrids be also formed in trans using Rad51 like it was shown at telomeres? would transcription inhibition fully prevent hybrid accumulation in the mutant?

Minor comment: figure 4f seems to have been generated using data from 2 experiments. It is usually not recommended to perform statistical analyses on 2 data points. It would be preferable to repeat the assay a third time.

Reviewer #3

(Remarks to the Author)

RNA:DNA hybrids are a major source of genomic instability in all organisms, from bacteria to humans. It is widely believed that these structures act as replication fork barriers and interfere with the faithful transmission of genetic and epigenetic information. However, the mechanism by which these structures induce replication stress and DNA damage remains poorly understood. In this manuscript, Mangione and colleagues used a series of systematic screens to re-examine the interplay between RNA:DNA and genomic instability in budding yeast. Specifically, they used the *S. cerevisiae* deletion library to identify non-essential genes whose loss increases recombination at a YFP reporter cassette containing an R-loop forming gene. They identified 114 hyper-rec mutants that were further analyzed with a secondary genetic screen that measured recombination at YAT1, another R-loop forming reporter gene. The 39 yeast genes whose deletion increased recombination in both assays were further tested for their effect on global R-loop accumulation using the S9.6 antibody and slot blot analysis. Remarkably, they found that the abundance of RNA:DNA hybrids did not correlate with recombination frequency, as some of the deletions led to massive increases in RNA:DNA hybrid levels without increased genomic instability. Since many of these mutants had in common the persistence of ssDNA gaps or flaps after DNA replication, the authors propose an attractive model in which these gaps precede and promote R-loop formation. In these mutants, increased levels of RNA:DNA hybrids would therefore be a consequence and not the cause of replication-induced discontinuities in DNA. They also used a quantitative and strand-specific DRIP-seq approach to show that R-loops form on the coding strand of active genes, consistent with the view that replication-borne discontinuities promote the formation of co-transcriptional R-loops. Overall, the data are of high quality and their interpretation is sound. This manuscript sheds new light on the interplay between R-loops and genome instability. In principle, it should be of interest to a wide audience. However, the authors need to address the issues listed below to strengthen their argument.

Specific issues:

1. The title needs to be more specific. As it is, it could be read as if all RNA:DNA hybrids result from pre-existing DNA lesions, which is not the main conclusion of the paper.
2. A large body of evidence suggests that frontal collisions between RNA and DNA polymerases are more deleterious than codirectional conflicts. What is the orientation of the *LYS2* and *YAT1* genes relative to the nearest origin of replication? Would the results of the screens be different if these genes were inverted?
3. Page 3, second paragraph, the authors briefly compare the result of their screen to another SSA screen, but do not really elaborate on this difference. More generally, how do the results of the HR screens compare to previous studies?
4. The model proposed by the authors is that cotranscriptional R-loops are more likely to form when ssDNA gaps are present. In *rad27* mutants or other mutants with altered Okazaki fragment processing, these breaks should be more frequent on the lagging strand. In principle, this should lead to a bias in hybrid intensity depending on the position and orientation of the replication origins. The authors should use their very nice DRIP-seq datasets to address this possibility.

5. Although the in vitro and in vivo evidence supporting the view that DNA damage precedes R-loop formation in RAD27-AID cells is compelling, the authors would strengthen their case by addressing this sequence of events experimentally, e.g., by depleting Rad27 in G1 and monitoring the formation of RNA:DNA hybrids by slot blot or DRIP-qPCR upon release of cells into the cell cycle.

6. Rad27 and most of the factors studied here (with the exception of Rad57 and Ctf18) have altered Okazaki fragment processing, which may artifactually increase S9.6 signals. To formally rule out this possibility, the authors should monitor R-loop levels in a mutant that accumulates DNA breaks independently of Okazaki fragment processing, such as the rad3-102 allele of TFIIF.

Version 1:

Reviewer comments:

Reviewer #1

(Remarks to the Author)

The authors addressed most of my comments adequately. As a new addition, they show that "Similarly, in human cells, while FEN1 inhibition increased gH2AX DSB repair foci, in line with previous studies (39,40), their accumulation was still observed upon DNA:RNA hybrid removal through RNase H1 over-expression (Fig. 4g; Supplementary Fig. 6d)". I could identify convincing evidence for RNase H1 over-expression in these data panels, but not for DNA:RNA hybrid removal. Adding evidence for the latter for the conditions shown in new Fig. 4g would strengthen this result significantly and justify the conclusions drawn from it. Apart from this missing control, I endorse publication in Nature Communications and congratulate the authors on their findings.

Reviewer #2

(Remarks to the Author)

The authors have addressed my comments appropriately. I recommend the article for publication.

Reviewer #3

(Remarks to the Author)

The authors have done a great job revising their manuscript. In my opinion, it is now suitable for publication.

Reviewer #1

Mangione et al. report on DNA:RNA hybrids forming at DNA replication-associated discontinuities of the DNA double-helix, such as unprocessed DNA flap structures and unligated Okazaki fragment. They conclude that DNA:RNA hybrids do not only form prior to DNA lesions, causing genome instability (“pre-lesion” hybrids), but commonly also as consequence of DNA lesions (“post-lesion” hybrids), without necessarily enhancing genetic instability. This is an important finding, given that DNA:RNA hybrids and especially R-loops have been investigated primarily as DNA damage-inducing structures. The genetic screens and validation experiments seem overall well conducted and the paper is very well written. If “post-lesion” hybrids are indeed a frequent and common consequence of DNA discontinuities, the manuscript provides an important counterpoint to the prevailing view on R-loop-induced genotoxicity.

We acknowledge Reviewer #1 for this very positive assessment of our work and for their suggestions for improvement. We have now included additional experiments and analyses to address the points they raised, as detailed below.

Main points:

1) The manuscript focusses primarily on yeast, but the results do suggest that human FEN1 has a similar role as Rad27 in suppressing flap-induced hybrids. However, only a single experiment with FEN1 is shown with limited specificity controls. In light of the broad interest in DNA:RNA hybrids and their role in human disease, it would be important to strengthen this point and extend the analysis of FEN1 inhibition.

Specifically:

- a. Include specificity controls for RHINO-mediated detection of hybrids (e.g. a binding-deficient mutant, as done by Martin et al., in reference 40), provide representative images, and validate FEN1 inhibition.
- b. Include an orthogonal approach (e.g. DRIP-qPCR/seq) and test if DNA discontinuities upon FEN1 inactivation precede and are sufficient for hybrid formation.
- c. Include a second FEN1 or FEN1/EXO1 inhibitor (e.g. LNT1).
- d. Test if post-lesion hybrids in mammalian cells impact genome stability.

In this study, the utilization of the yeast model allowed us to dissect the mechanistic relationships between hybrids and DNA damage upon Okazaki fragment processing defects (Fig. 2-3), and to generalize our observations to several other mutant situations (Fig. 4). Although the relevance of our findings is also worth investigating in human cells, we do not believe it would be possible to achieve a similar level of systematicity (e.g. screens, DRIP-seq) and mechanistic refinement within the timeframe of a revision. However, we have completed our analyses of FEN1 inhibition by including additional control and experiments:

- We now provide representative images of RHINO-mediated detection of hybrids in control and treated cells (Novel **Supplementary Fig. 2a**). We also feature RHINO specificity controls, *i.e.* binding deficient mutants (W34A and KK59/60AA, as in Martin et al, our reference #41), revealing the absence of detectable focal signals in control and treated cells (Novel **Supplementary Fig. 2b**).
- The biological activity of both inhibitors was confirmed by assessing DNA damage formation (*i.e.* γ H2AX; novel **Fig. 4g** and **Supplementary Fig. 6d**), in line with previous reports (our references #39-40).

- We have included a second FEN1 inhibitor (LNT1, our reference #40). Treatment with both inhibitors (SC13 and LNT1) similarly leads to hybrid accumulation (**Fig. 1j**; novel **Supplementary Fig. 2a, c**) and DNA damage formation (novel **Fig. 4g** and **Supplementary Fig. 6d**).

- We also report that DNA damage (γ H2AX) induction by both FEN1 inhibitors is still detectable upon hybrid removal through RNase H1 overexpression (novel **Fig. 4g**). This result indicates that “*post-lesion hybrids accumulating upon OF processing defects do not contribute measurably to intrinsic genetic instability*” in human cells, as now stated in the **Results** section (p7).

39. He, L. *et al.* Targeting DNA Flap Endonuclease 1 to Impede Breast Cancer Progression. *EBioMedicine* **14**, 32–43 (2016).

40. Exell, J. C. *et al.* Cellularly active N-hydroxyurea FEN1 inhibitors block substrate entry to the active site. *Nat. Chem. Biol.* **12**, 815–821 (2016).

41. Martin, R. M., de Almeida, M. R., Gameiro, E. & de Almeida, S. F. Live-cell imaging unveils distinct R-loop populations with heterogeneous dynamics. *Nucleic Acids Res.* **51**, 11010–11023 (2023).

2) While the conclusion that hybrids can form as “post-lesion” hybrids seems overall well supported by the data, the functional implications are less clear. The authors found no evidence for enhanced recombination frequency or enhanced DNA damage measured by Rad52 foci, but it is hard to believe that such hybrids would be completely neutral for genome stability. Would such hybrids become genotoxic and lead to genetic instability in certain contexts? The authors should try to identify conditions to which “post-lesion” hybrids sensitize (either by a hypothesis-driven approach or genetic screens).

We agree with the reviewer that “post-lesion” hybrids might not be fully neutral for genome integrity. Indeed, the fact that Rad27-depleted cells did not exhibit enhanced genetic instability upon the formation of hybrids could be due to the existence of cellular pathways masking their deleterious outcome. To investigate this question, we have combined Rad27 depletion with the inactivation of *bona fide* hybrid-processing factors, revealing strong genetic interactions with the double inactivation of RNase H1 and RNase H2, on the one hand, and with *SEN1*^{SETX} inactivation, on the other hand (Novel **Supplementary Fig. 8d-e**), as now featured and discussed in the manuscript (**Discussion** section, p9):

“*It is noteworthy that hybrids forming at pre-existing DNA lesions do not, as such, detectably aggravate their recombinogenicity (Fig. 4). However, it is likely that the accumulation and genotoxicity of these hybrids is restricted through alternative mechanisms, ultimately preserving genome stability and cell fitness. Consistently, we found that removing simultaneously both RNase H1 and RNase H2 activities led to strict synthetic lethality in combination with Rad27 depletion (Supplementary Fig. 8d), in line with reported systematic screens⁸³. Similarly, inactivation of the Sen1 DNA:RNA hybrid helicase triggered strong growth defects in Rad27-depleted cells (Supplementary Fig. 8e).*”

83. Chang, E. Y.-C. *et al.* MRE11-RAD50-NBS1 promotes Fanconi Anemia R-loop suppression at transcription-replication conflicts. *Nat. Commun.* **10**, 4265 (2019).

Additional points:

1) The authors conclude that they are scoring mostly transcription-dependent hybrids (Fig. 2). In the DRIP-qPCR experiments, this could probably be tested directly by adding a transcription inhibitor to the + auxin conditions.

To formally establish that the observed hybrids are transcription-dependent, we performed two complementary experiments (as similarly answered to Reviewer #2). On the one hand, cells were treated with 1,10-phenanthroline to *globally* inhibit transcription, which led to a reduction in hybrid accumulation in both *wt* and *rad27* mutant cells, as assessed by dot blot assay (Novel **Supplementary**

Fig. 4b). Alternatively, we used a TetOFF reporter system in which the transcription of the *YAT1* hybrid-forming sequence can be *specifically* repressed by doxycyclin addition, prior to DRIP-qPCR analysis (Novel **Fig. 2k**). This assay revealed that the “*DNA:RNA hybrids detected in wt cells and accumulating in the absence of Rad27 are formed in a transcription-dependent manner*” (as now stated in the Results section, p5).

2) Examples from previous studies, where hybrids or R-loops were already observed as consequence of replication stress (references 25-28), could be discussed in the light of the new findings from the current study in more depth in the discussion section of the manuscript.

We now discuss that hybrids previously detected in distinct situations of replication stress could also arise from DNA damage or discontinuities (**Discussion** section, p8):

“In the same line, DNA discontinuities, SSBs or gaps could also contribute to the reported accumulation of hybrids scored along with (i) replication fork pausing, as caused by dNTP deprivation upon HU treatment^{26–28} or fork barrier activity⁵⁴, and (ii) replisome instability in situations of head-on transcription-replication conflicts, as triggered at dedicated reporter loci in bacteria⁷¹, yeast⁷² and human²⁵.”

25. Hamperl, S., Bocek, M. J., Saldivar, J. C., Swigut, T. & Cimprich, K. A. Transcription-Replication Conflict Orientation Modulates R-Loop Levels and Activates Distinct DNA Damage Responses. *Cell* 170, 774-786.e19 (2017).

26. Heuzé, J. et al. RNase H2 degrades toxic RNA:DNA hybrids behind stalled forks to promote replication restart. *EMBO J.* 42, e113104 (2023).

27. Teloni, F. et al. Efficient Pre-mRNA Cleavage Prevents Replication-Stress-Associated Genome Instability. *Mol. Cell* 73, 670-683.e12 (2019).

28. Andrs, M. et al. Excessive reactive oxygen species induce transcription-dependent replication stress. *Nat. Commun.* 14, 1791 (2023).

54. Aiello, U. et al. Sen1 is a key regulator of transcription-driven conflicts. *Mol. Cell* 82, 2952-2966.e6 (2022).

71. Lang, K. S. et al. Replication-Transcription Conflicts Generate R-Loops that Orchestrate Bacterial Stress Survival and Pathogenesis. *Cell* 170, 787-799.e18 (2017).

72. Zardoni, L. et al. Elongating RNA polymerase II and RNA:DNA hybrids hinder fork progression and gene expression at sites of head-on replication-transcription collisions. *Nucleic Acids Res.* 49, 12769–12784 (2021).

Reviewer #2

In this manuscript, the authors bring evidence that DNA damage triggers the formation of genic DNA-RNA hybrids. Yeast is the model organism. The authors first looked for genes whose deletion is responsible for both recombination events and R-loop accumulation. Several factors were found but the authors particularly highlight 2 genes involved in Okazaki fragment processing (*rad27* (human FEN1) and *cdc9*). The authors claim that the R-loop accumulation in these mutants is not directly responsible for the recombinogenic events because these would mostly be occurring due to defective OF processing. Using these screens as the premise for the study, they rather hypothesize that the hybrid accumulation is triggered by the lesions occurring following OF processing defects. They next focus on *Rad27* for the study of the R-loop formation timing using AID system. The article is clear and well-written. The study is well conducted with a mix of cell work and biochemical approaches, and the evidence brought by the authors is convincing.

I have only a few suggestions and questions.

We would like to acknowledge Reviewer #2 for their enthusiastic evaluation of our study. We have now answered to their comments by providing additional experiments or analyses, as indicated below.

I would suggest illustrating graph 1J with representative images of the hybrid accumulation using the RHINO probes.

We now provide representative images of RHINO-expressing cells to illustrate hybrid accumulation upon FEN1 inhibition (Novel **Supplementary Fig 2a**). We also feature RHINO specificity controls, *i.e.* binding deficient mutants, revealing the absence of detectable focal signals in control and treated cells (Novel **Supplementary Fig. 2b**; see also Answer to Reviewer #1).

In Figure 2, the authors show the replication-dependency of hybrid formation by synchronizing the cells in G1. Replication can trigger DNA damage in other ways than by defective OF processing. Can Emetine which prevents OF formation be used to strengthen the conclusion that hybrid formation is specifically triggered by OF processing defects?

While emetine has been used in mammalian cells to interfere with OF formation and lagging strand synthesis, recent evidence indicates that its effects on replication are indirectly mediated through global inhibition of protein translation (Lukac et al, 2022; PMID: 36260751), confounding the interpretation of such experiments. We agree that we cannot formally exclude that replicative stress arising in OF processing mutants could participate to hybrid formation, although the observed genetic interactions between *rad27*, *exo1* and *cdc9* mutant hybrid-accumulating phenotypes (**Fig. 3h,j,k**) support a model in which OF-borne discontinuities are the main trigger of hybrid accumulation. This is now clearly stated in the text (**Results**, p6):

*“While we cannot exclude that replicative stress or other replication-dependent damage arising in these genetic contexts could participate to hybrid formation, these data indicate that the accumulation of discontinuities within neosynthesized lagging strands, *i.e.* unprocessed flaps, or nicks between unligated OF termini, is sufficient to enhance the formation of DNA:RNA hybrids at hybrid-prone transcribed regions.”*

The authors show that hybrid accumulation depends on transcription and occurs because of flap accumulation. However, they also show that transcription is not increased upon *rad27* deletion which could be counter-intuitive. The authors discuss this in the discussion section but do not mention a possible formation in trans. Can these hybrids be also formed in trans using Rad51 like it was shown at telomeres?

As pointed out, several lines of evidence have implicated strand exchange activities (*e.g.* Rad51) in the formation of DNA:RNA hybrids, both at telomeres (Feretzaki et al., 2020; our reference #75) and at other chromosomal loci (Wahba et al., 2013, Aronica et al, 2016; our references #76-77). To investigate whether Rad51 facilitates DNA:RNA hybrid formation upon defective Okazaki fragment processing, we performed DRIP-qPCR upon Rad27 depletion in *rad51Δ* cells. This experiment revealed that Rad51 does not detectably contribute to hybrid formation in the absence of Rad27, as now featured (Novel **Supplementary Fig. 8b**) and commented in the manuscript (**Discussion**, p9):

*“In this situation, RNA invasion would not necessarily require strand exchange activities, which were shown to contribute to hybrid formation in vivo⁷⁵⁻⁷⁷. Consistently, we found that inactivation of Rad51 did not impact hybrid accumulation upon OF processing defects (**Supplementary Fig. 8b**).”*

75. Feretzaki, M. et al. RAD51-dependent recruitment of TERRA lncRNA to telomeres through R-loops. *Nature* 587, 303–308 (2020).

76. Wahba, L., Gore, S. K. & Koshland, D. The homologous recombination machinery modulates the formation of RNA-DNA hybrids and associated chromosome instability. *Elife* 2, e00505 (2013).

77. Aronica, L. et al. The spliceosome-associated protein Nrl1 suppresses homologous recombination-dependent R-loop formation in fission yeast. *Nucleic Acids Res* 44, 1703–17 (2016).

Would transcription inhibition fully prevent hybrid accumulation in the mutant?

As similarly answered to Reviewer #1, to formally establish that the observed hybrids are transcription-dependent, we performed two complementary experiments. On the one hand, cells were treated with 1,10-phenanthroline to *globally* inhibit transcription, prior to hybrid detection by dot blot (Novel **Supplementary Fig. 4b**). On the other hand, we used a TetOFF reporter system in which the transcription of the *YAT1* hybrid-forming sequence can be *specifically* repressed by doxycyclin addition, prior to DRIP-qPCR analysis (Novel **Fig. 2k**). Both assays revealed that the “*DNA:RNA hybrids detected in wt cells and accumulating in the absence of Rad27 are formed in a transcription-dependent manner*” (as now stated in the **Results** section, p5).

Minor comment: figure 4f seems to have been generated using data from 2 experiments. It is usually not recommended to perform statistical analyses on 2 data points. It would be preferable to repeat the assay a third time.

Three replicates have now been performed for all Rad52 foci analyses and included in the corresponding figures (Novel **Fig. 4f** and **4k**). The conclusions of these experiments remain unchanged.

Reviewer #3

RNA:DNA hybrids are a major source of genomic instability in all organisms, from bacteria to humans. It is widely believed that these structures act as replication fork barriers and interfere with the faithful transmission of genetic and epigenetic information. However, the mechanism by which these structures induce replication stress and DNA damage remains poorly understood. In this manuscript, Mangione and colleagues used a series of systematic screens to re-examine the interplay between RNA:DNA and genomic instability in budding yeast. Specifically, they used the *S. cerevisiae* deletion library to identify non-essential genes whose loss increases recombination at a YFP reporter cassette containing an R-loop forming gene. They identified 114 hyper-rec mutants that were further analyzed with a secondary genetic screen that measured recombination at *YAT1*, another R-loop forming reporter gene. The 39 yeast genes whose deletion increased recombination in both assays were further tested for their effect on global R-loop accumulation using the S9.6 antibody and slot blot analysis. Remarkably, they found that the abundance of RNA:DNA hybrids did not correlate with recombination frequency, as some of the deletions led to massive increases in RNA:DNA hybrid levels without increased genomic instability. Since many of these mutants had in common the persistence of ssDNA gaps or flaps after DNA replication, the authors propose an attractive model in which these gaps precede and promote R-loop formation. In these mutants, increased levels of RNA:DNA hybrids would therefore be a consequence and not the cause of replication-induced discontinuities in DNA. They also used a quantitative and strand-specific DRIP-seq approach to show that R-loops form on the coding strand of active genes, consistent with the view that replication-borne discontinuities promote the formation of co-transcriptional R-loops. Overall, the data are of high quality and their interpretation is sound. This manuscript sheds new light on the interplay between R-loops and genome instability. In principle, it should be of interest to a wide audience. However, the authors need to address the issues listed below to strengthen their argument.

We acknowledge Reviewer #3 for their interest and positive evaluation of our work. We have now incorporated additional experiments and analyses to address the points raised, as detailed below.

Specific issues:

1. The title needs to be more specific. As it is, it could be read as if all RNA:DNA hybrids result from pre-existing DNA lesions, which is not the main conclusion of the paper.

To indicate that not all hybrids arise from DNA lesions, we have now modified the title as follows: “DNA lesions can frequently precede DNA:RNA hybrid accumulation”.

Indeed, our screening approaches have revealed that situations where lesions precede hybrid formation (“post-lesion” hybrids) are more frequent than those where hybrids are the source of DNA damage (“pre-lesion” hybrids).

2. A large body of evidence suggests that frontal collisions between RNA and DNA polymerases are more deleterious than codirectional conflicts. What is the orientation of the *LYS2* and *YAT1* genes relative to the nearest origin of replication? Would the results of the screens be different if these genes were inverted?

Within the reporters used for screens I and II, both “*LYS2* and *YAT1* genes are positioned opposite to the unique plasmid bidirectional replication origin (ARS, red dot), and can be replicated in both directions with respect to the orientation of transcription”, as now clearly stated in **Figure 1 legend**.

To investigate whether the position of reporter genes relative to the ARS influences the outcome of these assays, we further modified the *YAT1* construct to place the transcribed unit in either co-directional (CD) or head-on (HO) orientation relative to the ARS (**Fig. R1a**, ‘CD’ and ‘HO’), and performed hyper-recombination assays in *tho* and *rad27* mutants. Overall, both mutants scored as hyper-recombinant in this assay, regardless of the orientation of replication (**Fig. R1b**). However, addressing this question in a systematic manner would require to perform the complete screens with these novel reporters, a perspective for future studies.

Fig. R1 – Loss of the THO complex or Rad27 similarly triggers hyper-recombination on transcribed reporters, regardless of their orientation with respect to replication. **a**, Principle of the reporter constructs used here. Transcription of the *YAT1* hybrid-prone sequence is induced by galactose and recombination is scored by measuring the fraction of Leu⁺ prototrophs, arising from direct-repeat recombination. The three depicted constructs differ by the position of the plasmid autonomous replicating sequence (ARS), which dictates the orientation of the transcribed unit with respect to replication, and thereby the predominant type of transcription-replication conflict: unbiased, co-directional (CD), head-on (HO). **b**, Recombination frequencies calculated as described in Methods (fraction of Leu⁺ prototrophs, x10⁴; n = 14 (unbiased) and 34 (CD and HO) from at least 3 independent experiments).

3. Page 3, second paragraph, the authors briefly compare the result of their screen to another SSA screen, but do not really elaborate on this difference. More generally, how do the results of the HR screens compare to previous studies?

We now provide a detailed comparison of our results with two available SSA screen datasets (Segura-Wang et al. 2017, Novarina et al. 2020; our references #36,38). Remarkably, this analysis revealed a limited overlap between our hyper-rec subset and those identified in these other studies (see our modified **Supplementary Table 1**). As we now point out in the **Results** section (p3), this difference could stem from the specific usage of hybrid-prone sequences in our reporter designs: *“More generally, these successive screens identified a specific set of hyper-recombinant mutants compared to published SSA screens based on reporters that do not carry any R-loop forming sequence^{36,38} (Supplementary Table 1), thus highlighting our potential to hereby capture R-loop dependent events.”*

36. Novarina, D. et al. A Genome-Wide Screen for Genes Affecting Spontaneous Direct-Repeat Recombination in *Saccharomyces cerevisiae*. *G3 Bethesda Md* 10, 1853–1867 (2020).

38. Segura-Wang, M., Onishi-Seebacher, M., Stütz, A. M., Mardin, B. R. & Korbel, J. O. Systematic Identification of Determinants for Single-Strand Annealing-Mediated Deletion Formation in *Saccharomyces cerevisiae*. *G3 Bethesda Md* 7, 3269–3279 (2017).

4. The model proposed by the authors is that cotranscriptional R-loops are more likely to form when ssDNA gaps are present. In *rad27* mutants or other mutants with altered Okazaki fragment processing, these breaks should be more frequent on the lagging strand. In principle, this should lead to a bias in hybrid intensity depending on the position and orientation of the replication origins. The authors should use their very nice DRIP-seq datasets to address this possibility.

Our ssDRIP-seq procedure does not intrinsically discriminate signals from the two sister chromatids, and thereby cannot assess whether hybrids actually accumulate on the lagging strand upon defective OF processing. We have however analyzed whether replication orientation would impact hybrid accumulation in this situation, in the aim of investigating the mechanisms at play. For this purpose, we intersected our ssDRIP-seq dataset with available OK-seq data (McGuffee, Smith & Whitehouse, 2013; our reference #91) to compare hybrid signals for genes transcribed in CD or HO orientations relative to replication (see **Methods**, p27). This analysis revealed that both categories exhibit hybrid accumulation upon *Rad27* inactivation, a phenotype slightly more pronounced for CD genes (Novel **Supplementary Fig. 8c**). These observations are consistent with our speculative models predicting hybrid formation in both orientations, either through nascent RNA annealing or RNA polymerase pausing at lesions, as now discussed (**Discussion**, p9) and illustrated (Novel **Supplementary Fig. 8a**).

91. McGuffee, S. R., Smith, D. J. & Whitehouse, I. Quantitative, genome-wide analysis of eukaryotic replication initiation and termination. *Mol. Cell* 50, 123–135 (2013).

5. Although the in vitro and in vivo evidence supporting the view that DNA damage precedes R-loop formation in *RAD27-AID* cells is compelling, the authors would strengthen their case by addressing this sequence of events experimentally, e.g., by depleting *Rad27* in G1 and monitoring the formation of RNA:DNA hybrids by slot blot or DRIP-qPCR upon release of cells into the cell cycle.

As suggested, we released alpha-factor arrested cells into the cell cycle and monitored hybrid levels in early S and late S/G2 phases (30 min and 40 min after release, respectively; see our flow cytometry analysis in novel **Supplementary Fig. 4e**). As mentioned in the Results section (p5), *“DRIP-qPCR analysis revealed that the hybrid accumulation caused by *Rad27* loss coincided with S phase entry, with a decline in the late S/G2 phase”* (Novel **Supplementary Fig. 4f**). These observations further

support our model in which DNA discontinuities caused by defective OF processing precede hybrid accumulation.

6. Rad27 and most of the factors studied here (with the exception of Rad57 and Ctf18) have altered Okazaki fragment processing, which may artifactually increase S9.6 signals. To formally rule out this possibility, the authors should monitor R-loop levels in a mutant that accumulates DNA breaks independently of Okazaki fragment processing, such as the *rad3-102* allele of TFIIH.

Following this interesting suggestion, and based on previous observations that *rad3-102* mutant cells accumulate recombinogenic intermediates by failing to process UV-induced damage (Moriel-Carretero et al. 2010; PMID:20227372), we analyzed DNA:RNA hybrid levels in *wt* and mutant cells following UV irradiation. However, we were unable to detect an increase in hybrid signals in *rad3-102* cells, possibly due to the inaccessibility of the SSB to RNA invasion, consistent with TFIIH retention at damaged sites (as reported in the above study), or to local changes in RNA synthesis following UV damage (reviewed in Wang et al, 2023 PMID: 35731652).

To independently confirm that the S9.6 signals detected in OF processing mutants do not arise from OF RNA primers, we used two alternative strategies to inhibit transcription in the absence of Rad27 (as similarly answered to Reviewers #1 and #2). On the one hand, cells were treated with 1,10-phenanthroline to inhibit transcription, prior to dot blot analysis (Novel **Supplementary Fig. 4b**). On the other hand, we used a TetOFF reporter system in which the transcription of the *YAT1* hybrid-forming sequence can be repressed by doxycyclin addition, prior to DRIP-qPCR (Novel **Fig. 2k**). Both assays revealed that the “DNA:RNA hybrids detected in *wt* cells and accumulating in the absence of Rad27 are formed in a transcription-dependent manner” (as now stated in the **Results** section, p5).

Together with our results obtained with *rad57* Δ and *ctf18* Δ mutants, which do not impact OF processing, these data strongly support that S9.6 signals detected in our DRIP and dot blot assays do not stem from OF RNA primers.

Reviewer #1

The authors addressed most of my comments adequately. As a new addition, they show that "Similarly, in human cells, while FEN1 inhibition increased gH2AX DSB repair foci, in line with previous studies (39,40), their accumulation was still observed upon DNA:RNA hybrid removal through RNase H1 over-expression (Fig. 4g; Supplementary Fig. 6d)". I could identify convincing evidence for RNase H1 over-expression in these data panels, but not for DNA:RNA hybrid removal. Adding evidence for the latter for the conditions shown in new Fig. 4g would strengthen this result significantly and justify the conclusions drawn from it. Apart from this missing control, I endorse publication in Nature Communications and congratulate the authors on their findings.

We acknowledge the reviewer for their positive assessment. To address this remaining issue, we now provide representative images of RHINO-mediated detection of DNA:RNA hybrids in MCF7 cells, either control or treated with FEN1 inhibitors, and further expressing either wt or catalytic dead RNase H1 (Novel **Supplementary Figure 6e**). This control experiment revealed that wt RNase H1 over-expression led to a complete suppression of hybrid signals observed both in control and treated cells. Altogether, our observations thus fully support the conclusion that the genetic instability triggered by FEN1 inhibition in human cells (Fig. 4g) is not detectably caused by hybrid accumulation.

Reviewer #2

The authors have addressed my comments appropriately.
I recommend the article for publication.

We acknowledge the reviewer for their positive evaluation.

Reviewer #3

The authors have done a great job revising their manuscript. In my opinion, it is now suitable for publication.

We acknowledge the reviewer for this enthusiastic assessment.